# Spinal cord precursors utilize neural crest cell mechanisms to generate hybrid peripheral myelinating glia

Laura Fontenas, Sarah Kucenas*

Department of Biology, University of Virginia, Charlottesville, United States

**Abstract** During development, oligodendrocytes and Schwann cells myelinate central and peripheral nervous system axons, respectively, while motor exit point (MEP) glia are neural tube-derived, peripheral glia that myelinate axonal territory between these populations at MEP transition zones. From which specific neural tube precursors MEP glia are specified, and how they exit the neural tube to migrate onto peripheral motor axons, remain largely unknown. Here, using zebrafish, we found that MEP glia arise from lateral floor plate precursors and require *foxd3* to delaminate and exit the spinal cord. Additionally, we show that similar to Schwann cells, MEP glial development depends on axonally derived *neuregulin1*. Finally, our data demonstrate that overexpressing axonal cues is sufficient to generate additional MEP glia in the spinal cord. Overall, these studies provide new insight into how a novel population of hybrid, peripheral myelinating glia are generated from neural tube precursors and migrate into the periphery.

**\*For correspondence:**
sk4ub@virginia.edu

**Competing interests:** The authors declare that no competing interests exist.

## Introduction

The central and peripheral nervous systems (CNS and PNS, respectively) are two distinct yet connected compartments characterized by molecularly distinct cellular components that communicate to allow information to travel from the CNS to the periphery, and vice versa. Spinal motor nerves, which are composed of motor axons and their associated ensheathing glia, are essential to relay action potentials from the spinal cord to peripheral targets such as muscle. Motor exit point (MEP) transition zones (TZ) are specialized structures where motor axons exit the CNS, and where centrally and peripherally derived myelinating glia meet to myelinate the same axons (*Fontenas and Kucenas, 2018*; *Fraher, 1992*; *Fraher, 2002*). Until recently, TZs were thought to be selectively permeable to only axons, and the establishment of the territories occupied by glial cells remained poorly described and not well understood. Recent work now demonstrates that MEP TZs are occupied by dynamic glial cells and are precisely regulated over the course of nervous system development (*Coulpier et al., 2010*; *Fontenas et al., 2019*; *Kucenas et al., 2008b*; *Kucenas et al., 2009*; *Smith et al., 2016*; *Smith et al., 2014*; *Zhu et al., 2019*). CNS components such as oligodendrocyte (OL) lineage cells, motor neurons, and radial glia segregate and function in the spinal cord, whereas Schwann cells segregate and function in the PNS. However, other glial cells such as MEP glia, neural crest cells (NCCs), and perineurial glia, freely cross the MEP TZ as spinal motor nerves are being formed (*Kucenas et al., 2008b*; *Smith et al., 2014*; *Zhu et al., 2019*).

During nervous system development, neural tube precursors generate oligodendrocyte progenitor cells (OPC) that differentiate into OLs and ultimately myelinate axonal segments in the CNS, while Schwann cells originate from the neural crest and myelinate axons in the PNS (*Bergles and Richardson, 2015*; *Jessen and Mirsky, 2005*). Recently, we described MEP glia, a population of CNS-derived, peripheral glial cells that originate in the neural tube, exit through the MEP TZ, and eventually reside just outside of the spinal cord along motor nerve root axons, occupying territory between Schwann cells in the periphery and OLs in the spinal cord (*Smith et al., 2014*). In addition

to their central origin and ultimate peripheral fate, one of the unique features of MEP glia is their expression of both CNS and PNS identity markers, including *olig2* and *foxd3* (*Fontenas and Kucenas, 2018*; *Smith et al., 2014*). MEP glia express *foxd3* before they exit the neural tube, and its expression persists as they migrate into the periphery along motor axons (*Smith et al., 2014*). Although the function of *foxd3* in maintaining the fate of NCCs and repressing melanogenesis is known in chick, *Xenopus*, zebrafish, and mouse, its role in the ventral neural tube where MEP glia originate, remains unknown (*Dottori et al., 2001*; *Kos et al., 2001*; *Lister et al., 2006*; *Sasai et al., 2001*; *Teng et al., 2008*).

Once in the periphery, MEP glia tightly associate with spinal motor root axons and ultimately myelinate the most proximal axonal segments along the nerve, and therefore, participate in the fast conduction of action potentials from the CNS to the muscle (*Fontenas and Kucenas, 2018*; *Smith et al., 2014*). Previously, we showed that MEP glia also function to restrict highly migratory OPCs to the spinal cord (*Fontenas et al., 2019*; *Smith et al., 2014*). While we are beginning to appreciate the functional roles of MEP glia (*Fontenas et al., 2019*; *Smith et al., 2014*), we still lack a complete understanding of their development. MEP glia originate within the ventral neural tube, however, their precise origin and relationship to other CNS cell populations and peripheral Schwann cells, remain unclear (*Smith et al., 2014*).

Here, we describe the mechanisms that control MEP glial development. Using a combination of live imaging, clonal analysis, and genetic approaches in zebrafish, we show that MEP glia originate from *nkx2.2a*$^+$/*olig2*$^+$ radial glial precursors and require *foxd3* to delaminate from the lateral floor plate and exit the spinal cord via MEP TZs. We also show that MEP glia require axonal cues and classical NCC signaling cascades to migrate out of the spinal cord. We demonstrate that similar to Schwann cells, MEP glial development is driven by axonal Neuregulin signaling and that manipulating this signaling is sufficient to generate additional MEP glia from precursors in the post-embryonic spinal cord. Together, our results demonstrate that ventral spinal cord precursors employ NCC developmental mechanisms to generate hybrid, peripheral myelinating glia and bring a better understanding of MEP glial development and spinal motor nerve formation.

## Results

### Nkx2.2a$^+$/olig2$^+$/foxd3$^+$ neural tube radial glial precursors give rise to MEP glia

Although all peripheral myelinating glia were previously thought to originate from the neural crest, we recently described MEP glia, a population of peripheral spinal motor nerve root glia that are derived from *olig2*$^+$ ventral spinal cord precursors and exit the CNS to myelinate peripheral motor axons (*Figure 1A*; *Smith et al., 2014*). In our initial study, we reported that MEP glia originate from *olig2*$^+$ precursors in the spinal cord and like perineurial glia, exit the CNS through MEP TZs at approximately 48 hr post-fertilization (hpf). However, the precise mechanisms that govern their development remained uninvestigated (*Kucenas et al., 2008b*; *Smith et al., 2014*).

The vertebrate embryo neural tube is organized according to the expression of region-specific transcription factors along the antero-posterior and dorso-ventral axis in a spatially and temporally controlled manner. Precursor cells produce different cell types in a stereotypical sequence and the fate of the progeny is determined by when and where they were born (*Briscoe et al., 2000*). For example, distinct *olig2*-expressing neural precursors in the pMN domain sequentially generate motor neurons and oligodendrocytes (*Fu et al., 2002*; *Ravanelli and Appel, 2015*; *Zhou et al., 2001*). *Nkx2.2a* precursors, which lie just ventral to the pMN domain, generate interneurons, OLs, and perineurial glia, a subpopulation of centrally-derived peripheral glia that form the perineurium along motor nerves (*Briscoe et al., 1999*; *Clark et al., 2014*; *Deska-Gauthier et al., 2020*; *Fu et al., 2002*; *Kucenas et al., 2008a*; *Kucenas et al., 2008b*; *Morris et al., 2017*; *Schäfer et al., 2007*; *Soula et al., 2001*).

To investigate the origin of MEP glia and characterize their molecular identity in the periphery, we used in vivo imaging in different combinations of zebrafish transgenic lines between 48 hpf and 4 days post-fertilization (dpf). To visualize and distinguish MEP glia from perineurial glia, we used the *foxd3:mcherry* gene trap line, where *foxd3* regulatory sequences label NCCs and MEP glia (*Fontenas and Kucenas, 2018*; *Hochgreb-Hägele and Bronner, 2013*; *Smith et al., 2014*), in

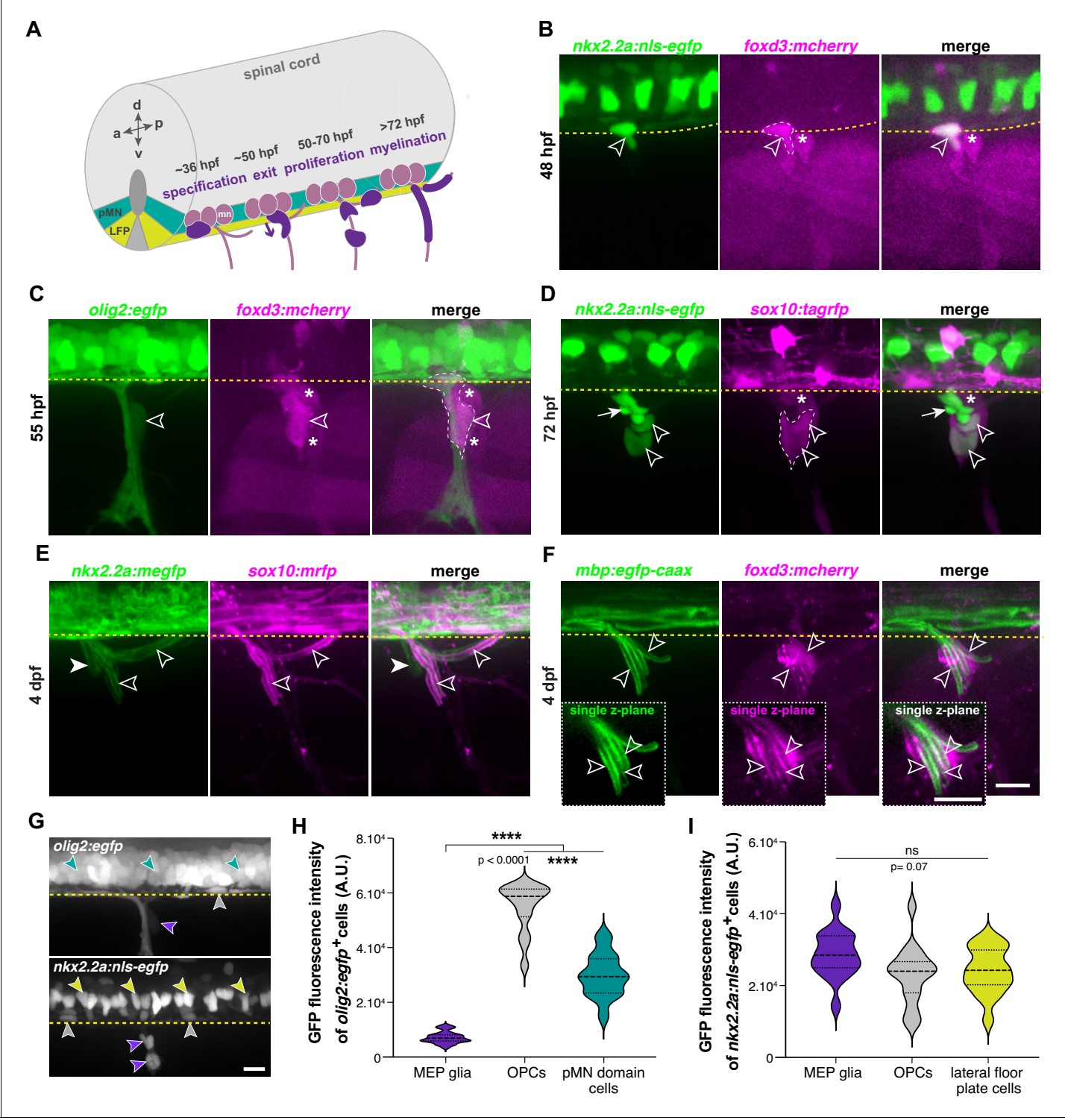

**Figure 1.** MEP glia are hybrid, centrally derived, peripheral myelinating glia. (A) Timeline of MEP glial development. MEP glia (purple) are specified in the ventral spinal cord (gray) after 36 hpf, exit through motor exit points at ~50 hpf, divide, and migrate to eventually initiate myelination of motor root axons (pink) starting at 72 hpf. mn: motorneuron, LFP: lateral floor plate. (B–F) Lateral views of the motor exit point showing (B) a nkx2.2a⁺/foxd3⁺ MEP glia (outlined arrowhead) exiting the spinal cord at the motor exit point at 48 hr post-fertilization (hpf); (C) an olig2⁺/foxd3⁺ MEP glia (outlined arrowhead) along motor nerve root axons at 55 hpf; (D) sox10⁺/nkx2.2a⁺ MEP glia (outlined arrowheads) and sox10⁻/nkx2.2a⁺ perineurial glia (arrow) at 72 hpf; (E) sox10⁺/nkx2.2a⁺ MEP glial sheaths (outlined arrowheads) and sox10⁻/nkx2.2a⁺ perineurial cells at 4 days post-fertilization dp (arrowhead) and (F) foxd3⁺ MEP glia making mbp⁺ myelin sheaths (outlined arrowheads) at 4 dpf. Insets show single z-plane images. (G) In olig2:egfp and nkx2.2a:nls-

*Figure 1 continued on next page*

*Figure 1 continued*

*egfp* larvae at 55 hpf, *olig2*$^+$ pMN domain cells (teal), OPCs (gray), and MEP glia (purple) are labeled, as are *nkx2.2a*$^+$ LFP cells (yellow), OPCs (gray) and MEP glia (purple). These images were used for fluorescence intensity measurement. (H) Violin plot of mean intensity of GFP fluorescence of *olig2:egfp*$^+$ cells at 55 hpf (OPCs: 54914 ± 1674 arbitrary units (A.U.), MEP glia: 7149 ± 394, pMN domain cells: 29934 ± 1481). (I) Violin plot of mean intensity of GFP fluorescence of *nkx2.2a:nls- egfp* cells at 55 hpf (OPCs: 22981 ± 2013, MEP glia: 28610 ± 1624, lateral floor plate cells: 24049 ± 1602). (H–I) (n = 28 MEP glia, n = 28 OPCs and n = 28 neural tube cells from seven embryos). Asterisks denote the dorsal root ganglion (DRG) and yellow dashed lines denote the edge of the spinal cord. Scale bar (B–G) 20 µm.

The online version of this article includes the following source data and figure supplement(s) for figure 1:

**Source data 1.** Source data for *Figure 1* .
**Figure supplement 1.** Development of *nkx2.2a*+MEP glia.
**Figure supplement 1—source data 1.** Source data for *Figure 1—figure supplement 1* .
**Figure supplement 2.** Fluorescence intensity over the timecourse of MEP glial development.
**Figure supplement 2—source data 1.** Source data for *Figure 1—figure supplement 2* .
**Figure supplement 3.** Transverse sections of the neural tube of *olig2:egfp;nkx2.2a:nls-mcherry* embryos at 24, 30, 36, 48, and 72 hpf showing *olig2*$^+$/*nkx2.2a*$^+$ cells (outlined arrowheads).

combination with the ventral spinal cord reporter lines *nkx2.2a:nls-egfp* and *nkx2.2a:mcerulean*, where *nkx2.2a* regulatory sequences label the p3 domain (lateral floor plate) of the ventral spinal cord and its derivatives, including perineurial glia (*Zhu et al., 2019*). We began our imaging at 48 hpf as this is when we first observe CNS-derived, peripheral glia exiting the neural tube. At 48 hpf in *nkx2.2a:nls-egfp;foxd3:mcherry* embryos, we observed that MEP glia express the lateral floor plate marker *nkx2.2a* (*Figure 1B* and *Figure 1—figure supplement 1A*), which we and others had not previously observed, while perineurial glia, which also exit the spinal cord at approximately 50 hpf, expressed only *nkx2.2a* (*Figure 1B* and *Figure 1—figure supplement 1A,B*; *Kucenas et al., 2008b*). From these observations, we conclude that *nkx2.2a* is a new marker for MEP glia.

To confirm that the CNS-derived *nkx2.2a*$^+$/*foxd3*$^+$ cells we observed and the *olig2*$^+$/*foxd3*$^+$ MEP glia that we previously described were the same cells, we imaged *foxd3:mcherry;olig2:egfp* embryos, where *olig2* regulatory sequences label motor neurons, oligodendrocyte lineage cells, and MEP glia (*Fontenas and Kucenas, 2018*; *Park et al., 2007*; *Smith et al., 2014*). Consistent with previous studies, we found that MEP glia transiently expressed *olig2* until approximately 55 hpf (*Figure 1C* and *Figure 1—figure supplement 1C*; *Fontenas and Kucenas, 2018*; *Lee et al., 2020*; *Smith et al., 2014*). In all of our live imaging from 48 hpf to 4 dpf, MEP glia were the only *nkx2.2a*$^+$/*foxd3*$^+$ or *olig2*$^+$/*foxd3*$^+$ cells we detected in both the CNS and the PNS.

We next sought to verify that the *foxd3*$^+$/*nkx2.2a*$^+$ motor nerve-associated glia that we observed were MEP glia and not perineurial glia (*Kucenas et al., 2008b*). To do this, we used the *sox10:tagrfp* transgenic line, where *sox10* regulatory sequences label glia, including MEP glia, Schwann cells, and OLs, but not perineurial glia (*Binari et al., 2013*; *Kucenas et al., 2008b*; *Zhu et al., 2019*). At 72 hpf in nkx2.2a:nls-egfp;sox10:tagrfp larvae, we observed both *nkx2.2a*$^+$/*sox10*$^+$ MEP glia and *nkx2.2a*$^+$/*sox10*$^-$ perineurial glia along motor nerve axons (*Figure 1D*). When we looked at 4 dpf in *nkx2.2a: megfp;sox10:mrfp* larvae, we observed two morphologically distinct cell types, with *nkx2.2a*$^+$/*sox10*$^+$ MEP glial membrane sheaths that were reminiscent of myelin internodes (*Almeida et al., 2011*; *Almeida et al., 2018*; *Fontenas and Kucenas, 2018*; *Preston et al., 2019*), and *nkx2.2a*$^+$/*sox10*$^-$ loose perineurial cell membranes (*Figure 1E*; *Binari et al., 2013*; *Kucenas et al., 2008b*). To confirm that these *nkx2.2a*$^+$/*sox10*$^+$ sheaths along motor root axons were the same MEP glial myelin sheaths that we described previously (*Smith et al., 2014*), we imaged *foxd3:mcherry;mbp:egfp-caax* larvae, where *mbpa* regulatory sequences drive expression of membrane-tethered eGFP in myelinating glia (*Almeida et al., 2011*). At 4 dpf, we observed *mbp*$^+$ myelin sheaths formed by *foxd3*$^+$ cells along motor root axons just outside of the spinal cord (*Figure 1F*), in a pattern consistent with the location and morphology of MEP glia. By examining various combinations of double transgenic embryos and larvae, we conclude that MEP glia are *foxd3*$^+$/*sox10*$^+$/*olig2*$^+$/*nkx2.2a*$^+$/*mbp*$^+$ neural-tube-derived, peripheral myelinating glia that express a unique combination of central and peripheral markers, and identify *nkx2.2a* as a new marker for MEP glia.

In our imaging, we detected *olig2* expression in MEP glia. However, GFP fluorescence dramatically diminished shortly after they exited from the spinal cord (*Figure 1C*). Quantification of the eGFP fluorescence intensity in 55 hpf *olig2:egfp* embryos showed that MEP glial levels of eGFP

were 4.18 times lower than $olig2^+$ cells of the pMN domain and 7.68 times lower than OPCs (*Figure 1G,H*). Additionally, in situ hybridization with a mRNA probe specific to *olig2* did not show $olig2^+$ cell bodies in the PNS near the MEP TZ at 48 or 72 hpf (*Figure 1—figure supplement 1D*), demonstrating that MEP glia turn off *olig2* before they exit the CNS. However, measurements of the eGFP fluorescence intensity in 55 hpf *nkx2.2a:nls-egfp* embryos revealed no statistically significant difference in expression between MEP glia, lateral floor plate cells, and OPCs (*Figure 1G,I*). To confirm these findings, we also measured GFP fluorescence levels over the course of MEP glial development in the CNS and PNS using *foxd3:mcherry;nkx2.2a:nls-egfp* and *foxd3:mcherry;olig2:egfp* embryos and larvae at 48, 50, 55, and 72 hpf (*Figure 1—figure supplement 2A–C*). We observed that while *nkx2.2a*-driven fluorescence of GFP remained stable after MEP glial exit from the spinal cord (*Figure 1—figure supplement 2A,C*), we observed a 2.5 fold-decrease in *olig2*-driven expression of GFP at these same stages (*Figure 1—figure supplement 2B,C*). Because MEP glia express low levels of *olig2* (*Figure 1C,G,H*) but high levels of *nkx2.2a* (*Figure 1B,D,G,I*), and because *olig2* and *nkx2.2a* are identity markers of distinct neural tube precursor domains, we hypothesized that MEP glia originate from the *nkx2.2a* lateral floor plate and not from the pMN domain as we initially described (*Smith et al., 2014*).

We therefore investigated the structure of ventral neural tube domains prior to and during MEP glial development. We imaged transverse sections of the developing neural tube in *olig2:egfp; nkx2.2a:nls-mcherry* embryos and larvae from 24 to 72 hpf, a time window that encompasses the specification and migration of MEP glia. We observed that the *olig2* pMN domain and the *nkx2.2a* lateral floor plate of the neural tube, which are distinct at 24 hpf, lose their defined separation at approximately 30 hpf and merge to give rise to one mixed domain by 48 hpf, a phenomenon that has previously been observed and described during OPC specification (*Figure 1—figure supplement 3*; *Kessaris et al., 2001*; *Kucenas et al., 2008a*; *Scott et al., 2020*; *Soula et al., 2001*; *Tsai et al., 2020*; *Xiong et al., 2013*). Because MEP glia appear to express both *olig2* and *nkx2.2a*, we hypothesized that similar to OPCs, MEP glia originate from a mixed domain.

To determine if MEP glia come from $nkx2.2a^+/olig2^+$ precursors, we analyzed the co-expression of *foxd3* with *olig2* and *nkx2.2a* in the ventral spinal cord in *foxd3:mcherry;olig2:egfp* and *foxd3: mcherry;nkx2.2a:mcerulean* embryos prior to MEP glial exit. At 48 hpf, we observed $foxd3^+$ MEP glia with weak expression of *olig2*, just ventral to the pMN domain (*Figure 2A*). In *foxd3:mcherry; nkx2.2a:mcerulean* embryos, we observed floor plate $foxd3^+/nkx2.2a^+$ MEP glia at 48 hpf (*Figure 2B*). In all of our imaging, we always detected one $foxd3^+$ cell in the ventral spinal cord per hemi-segment of the zebrafish trunk (*Figure 1B*, *Figure 2A,B*, *Figure 2—figure supplement 1*). Therefore, we hypothesized that the $olig2^+/foxd3^+$ MEP glia and the $nkx2.2a^+/foxd3^+$ MEP glia were the same cell. To confirm this, we examined single z-planes of transverse sections of the neural tube in *foxd3:mcherry;nkx2.2a:mcerulean;olig2:egfp* embryos at 48 hpf. Consistent with our hypothesis, we observed $olig2^+/nkx2.2a^+/foxd3^+$ MEP glia within the mixed *olig2/nkx2.2a* domain of the spinal cord at 48 hpf (*Figure 2C*). We also observed that the shape and the position of these cells next to the central canal of the spinal cord were reminiscent of radial glia (*Figure 2A–C*).

Development of the CNS requires supporting and instructive scaffold cells, and this function is widely attributed to radial glia (*Hartfuss et al., 2001*; *McDermott et al., 2005*). In the spinal cord, radial glia are the first glial cells distinguishable from the neuroepithelium. They can be identified morphologically by their long radial fibers extending toward the pial surface and their cell bodies located along the ventricular zone, as well as their expression of region-specific transcription factors (*Lyons et al., 2003*; *Ogawa et al., 2005*). In addition to their scaffolding role, radial glia also function as neural precursors in the CNS (*Fogarty et al., 2005*; *Johnson et al., 2016*; *Park et al., 2007*). To better visualize the cellular architecture of the developing spinal cord, we labeled radial glia using the Zrf1 antibody which detects Glial Fibrillary Acidic Protein (GFAP) (*Trevarrow et al., 1990*). We found that all ventral spinal cord $nkx2.2a^+$ cells that we examined were $GFAP^+$ (*Figure 2D*; *Park and Appel, 2003*). For this reason, we will refer to MEP glial progenitors as $nkx2.2a^+$ radial glial precursors (*Kim et al., 2008a*; *Kim et al., 2008b*). From these studies, we conclude that MEP glia are generated from lateral floor plate $nkx2.2a^+/olig2^+$ radial glial precursors, and maintain expression of both CNS and PNS markers once in the PNS.

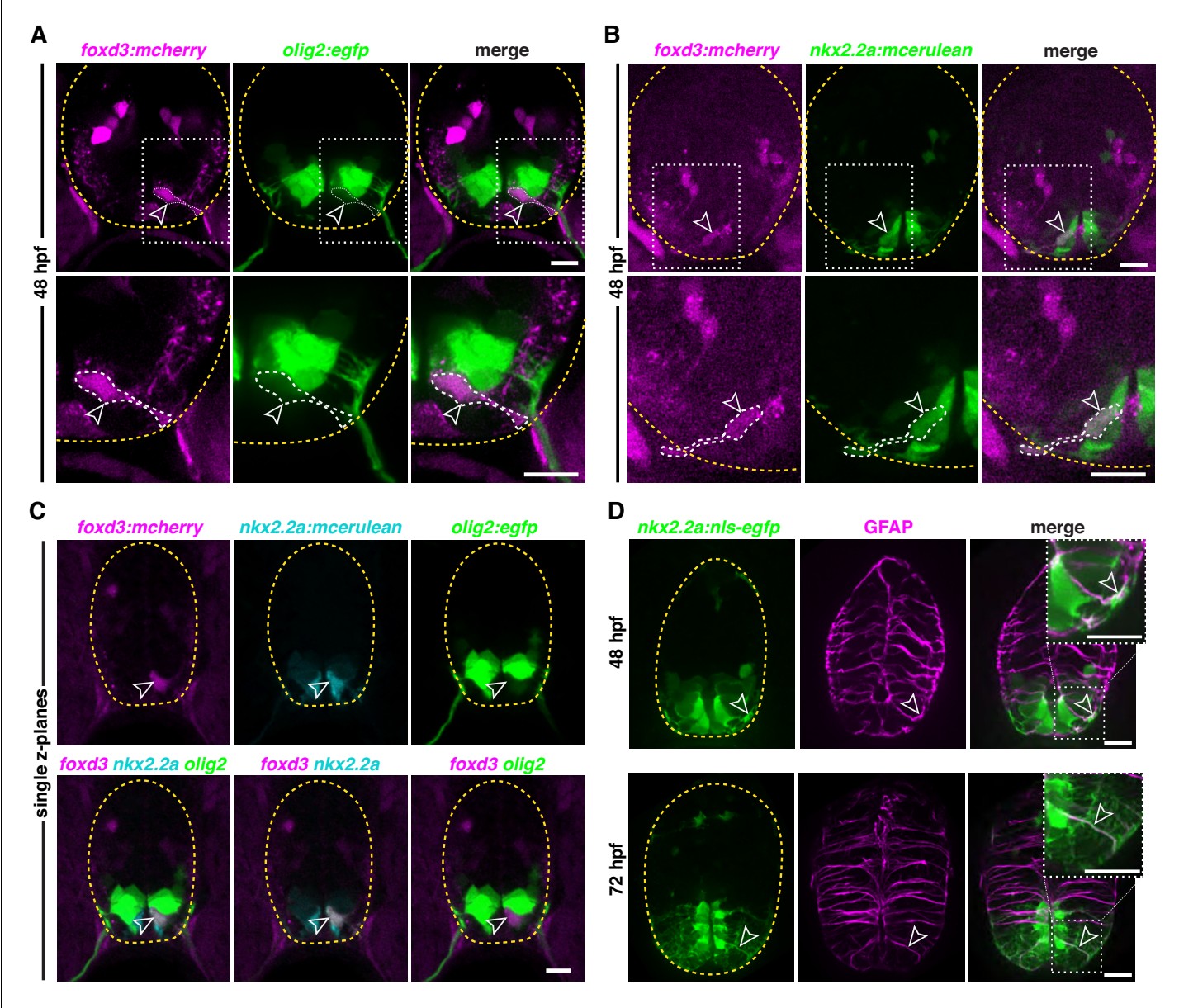

**Figure 2.** Ventral neural tube radial glial precursors give rise to MEP glia. (**A**) Transverse section of a *foxd3:mcherry;olig2:egfp* embryo at 48 hpf showing a *foxd3⁺/olig2⁺* MEP glial cell (outlined arrowhead) ventral to the *olig2* pMN domain in the spinal cord. (**B**) Transverse section of a *foxd3:mcherry;nkx2.2a:mcerulean* embryo at 48 hpf showing a *foxd3⁺/nkx2.2a⁺* MEP glial cell (outlined arrowhead). (**C**) Transverse section of a *foxd3:mcherry;nkx2.2a:mcerulean;olig2:egfp* embryo at 48 hpf showing a *foxd3⁺/nkx2.2a⁺/olig2⁺* triple positive MEP glial cell (outlined arrowhead) in the p3 domain of the neural tube, just ventral to the pMN domain. (**D**) Transverse section of a *nkx2.2a:nls-egfp* embryo showing *nkx2.2a⁺/GFAP⁺* radial glia (outlined arrowhead) at 48 hpf (top panel) and 72 hpf (bottom panel). Yellow dashed lines outline the edge of the spinal cord. Top right corner white boxes show higher magnification of bottom white boxes. Scale bar, (**A–D**) 10 μm.

The online version of this article includes the following figure supplement(s) for figure 2:

**Figure supplement 1.** Lateral view of the trunk of a *foxd3:mcherry;olig2:egfp* embryo showing one *foxd3⁺* MEP glial cell (outlined arrowhead) at the MEP TZ at 48 hpf.

## *Foxd3* is essential for MEP glial delamination and exit from the spinal cord

The fact that MEP glia originate from mixed, ventral spinal cord precursors wasn't surprising to us, given that *olig2* and *nkx2.2a* are expressed by CNS glia (*Kucenas et al., 2008a*; *Park et al., 2007*;

Soula et al., 2001; Sun et al., 2001; Tatsumi et al., 2018; Zhou et al., 2001). However, while there are many studies describing the roles of *foxd3* in peripheral tissue development, none have described a role for this gene in CNS glial development. To elucidate the role of *foxd3* in MEP glial development, we used the *foxd3:mcherry* gene trap zebrafish line, that when bred to homozygosity, creates a *foxd3* mutant (Hochgreb-Hägele and Bronner, 2013). While *foxd3:mcherry* heterozygous larvae label *foxd3*$^+$ MEP glia as well as neural crest-derived cells, including dorsal root ganglia (DRG) and Schwann cells (*Figure 1B–C*, *Figure 2A–C*, *Figure 1—figure supplement 1A* and *Figure 3A*), *foxd3:mcherry* homozygous larvae, which we will refer to as *foxd3*$^{-/-}$ larvae, lack these neural crest derivatives as well as MEP glia along spinal motor nerve axons (*Figure 3A*). When we time-lapse imaged *olig2:egfp;foxd3:mcherry;foxd3*$^{+/-}$ and $^{-/-}$ larvae from 48 to 72 hpf, we observed that *foxd3*$^+$ MEP glia migrated onto peripheral motor nerves in *foxd3*$^{+/-}$ siblings (*Figure 3A* and *Video 1*), while they stayed in the ventral spinal cord and failed to exit into the periphery in mutants (*Figure 3A* and *Video 2*). Because spinal cord OPCs developed at the expected developmental stage (*Figure 3A* and *Video 2*) and the morphology and body length of foxd3$^{-/-}$ larvae were comparable to their control siblings (*Figure 3B*), we ruled out the possibility that failure of MEP glial exit from the spinal cord was the consequence of developmental delay or perturbed neural tube patterning in *foxd3*$^{-/-}$ larvae. By 72 hpf, *foxd3*$^{+/-}$ siblings had peripheral MEP glia on 100% of their spinal motor nerve roots (*Figure 3A,C*). In contrast, we did not observe any MEP glia in the PNS of *foxd3*$^{-/-}$ larvae (*Figure 3A,C*). However, we did observe ectopic, peripheral OPCs along motor nerves in *foxd3*$^{-/-}$ larvae (*Figure 3A,D*), which is consistent with the absence of peripheral MEP glia (Fontenas et al., 2019; Smith et al., 2014).

One possible reason why we failed to observe MEP glia exit the spinal cord in *foxd3*$^{-/-}$ larvae is due to a failure of differentiation. To test this hypothesis, we performed in situ hybridization for the MEP glial marker *wif1* at 3 dpf (Smith et al., 2014). In *foxd3* heterozygous larvae, we observed *wif1*$^+$ MEP glia in the periphery (*Figure 3—figure supplement 1A*). In contrast, we detected the presence of *wif1*$^+$ MEP glia at the edge of the spinal cord near the MEP TZ, but not along the motor root, in *foxd3*$^{-/-}$ larvae (*Figure 3—figure supplement 1A*). Therefore, we conclude that MEP glia are present and can differentiate in the absence of Foxd3. To rule out the possibility that the *foxd3*$^+$ cells that we observed in the ventral spinal cord of *foxd3*$^{-/-}$ larvae were neurons, we performed immunohistochemistry with an anti-HuC antibody, which labels neurons, in *foxd3:mcherry;olig2:egfp* larvae at 72 hpf. Analysis of single z-plane confocal images revealed that the *olig2*$^+$/*foxd3*$^+$ cells we observed in the CNS of *foxd3*$^{-/-}$ larvae were HuC$^-$, and therefore, not neurons (*Figure 3—figure supplement 1B*).

Because *sox10* is essential for Schwann cell and OL specification and development (Britsch et al., 2001; Dutton et al., 2001; Kuhlbrodt et al., 1998; Takada et al., 2010) and is also expressed by MEP glia, we sought to determine whether MEP glia turn on Sox10 in *foxd3*$^{-/-}$ larvae. To do this, we performed immunostaining in *olig2:egfp;foxd3:mcherry* larvae using an antibody specific to zebrafish Sox10 (Binari et al., 2013). We found that similar to MEP glia in the periphery of control larvae, centrally located MEP glia in *foxd3*$^{-/-}$ larvae were Sox10$^+$ at 72 hpf (*Figure 3F*). To determine when Sox10 is turned on in MEP glia in relation to *foxd3*, we analyzed the expression of both markers in *foxd3:mcherry;sox10:nls-eos* embryos between 36 and 55 hpf. We tracked *foxd3*$^+$/*sox10*$^+$ MEP glia as early as we could detect them in the neural tube (between 44 and 48 hpf) by playing our time-lapse movies backwards and investigated which marker they expressed first. From our data, we observed that *foxd3* was detected first in 66.66% of the cells we examined, while *sox10* expression was detected first in 8.33% of the cells (*Figure 3G*). In 25% of the cells we analyzed, *foxd3* and *sox10* were detected simultaneously, likely because we did not detect the cells early enough in their development. From this data, we conclude that *sox10* expression in MEP glia is subsequent to, but independent of, *foxd3* expression. Taken together, we conclude that MEP glia are specified in *foxd3*$^{-/-}$ larvae.

To better understand the position of the stalled MEP glial cell in the CNS along the latero-medial axis, we imaged transverse sections of the spinal cord in *foxd3:mcherry;foxd3*$^{+/-}$ and mutant larvae at 72 hpf (*Figure 3E*). We found that while MEP glia were present along peripheral motor axons in heterozygous siblings, they were located in the floor plate in *foxd3*$^{-/-}$ larvae and had a morphology reminiscent of radial glia (*Figure 3E*). We conclude from these observations that MEP glia do not delaminate from the lateral floor plate in *foxd3* mutants.

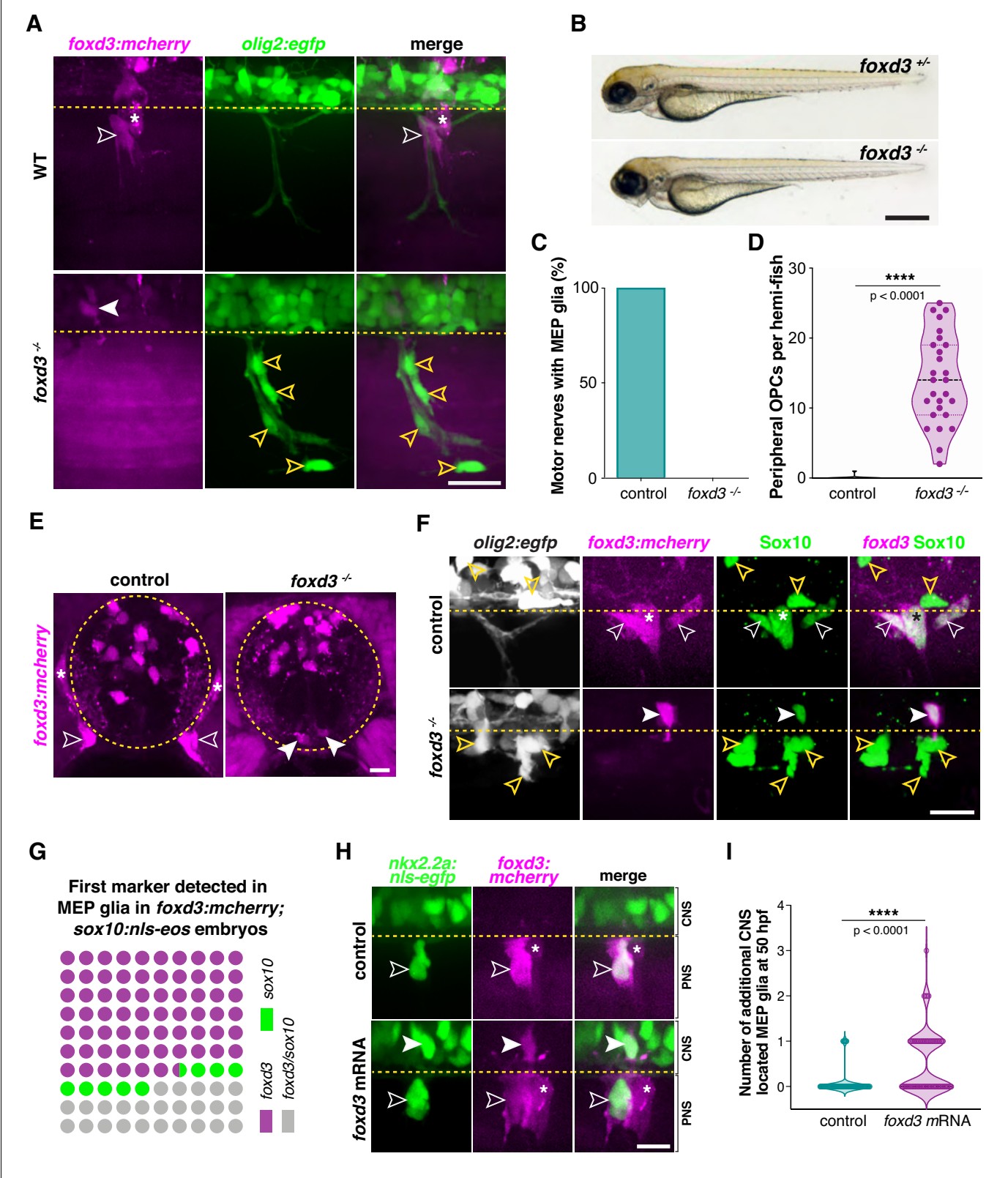

**Figure 3.** MEP glia require foxd3 to exit the spinal cord. (A) MEP TZ in a *foxd3:mcherry;olig2:egfp* larvae showing *foxd3⁺* MEP glia (white outlined arrowhead) along the motor root of a *foxd3⁺ᐟ⁻* control larva and stalled in the spinal cord of a *foxd3⁻ᐟ⁻* larva (white arrowhead) at 3 dpf. Note the presence of peripheral OPCs (yellow outlined arrowheads) along motor nerve axons in a *foxd3⁻ᐟ⁻* sibling lacking peripheral MEP glia. (B) Bright-field

*Figure 3 continued on next page*

*Figure 3 continued*

images of *foxd3*[+/-] and *foxd3*[-/-] siblings at 3 dpf reveal no developmental delay in *foxd3*[-/-] larvae. (C) Percentage of motor nerves with MEP glia in *foxd3*[+/-] (n = 150 nerves from 10 larvae) and *foxd3*[-/-] siblings (n = 270 nerves from 27 larvae) at 3 dpf. (D) Mean ± SEM of peripheral OPCs in *olig2:egfp; foxd3:mcherry;foxd3*[+/-] (0.07 ± 0.07, n = 15 larvae) and *foxd3*[-/-] (14.07 ± 1.2, n = 27 larvae) larvae at 3 dpf; p<0.0001. (E) Transverse sections of the spinal cord in *foxd3:mcherry;foxd3*[+/-] and *foxd3*[-/-] larvae showing MEP glia (outlined arrowheads) at the motor nerve root in a control sibling and MEP glia (arrowheads) in the lateral floor plate in a *foxd3* mutant at 72 hpf. (F) Immunohistochemistry showing Sox10[+]/*foxd3*[+] MEP glia (white outlined arrowheads) along motor nerve root axons in a *foxd3*[+/-] control embryo, and a Sox10[+]/*foxd3*[+] MEP glial cell (white arrowhead) stalled in the spinal cord of a *foxd3*[-/-] larvae at 3 dpf. Yellow outlined arrowheads show OPCs in the spinal cord in a control larva and along peripheral axons in a *foxd3*[-/-] larva. Asterisks denote the DRG in control larvae. Note the absence of DRG in *foxd3* mutants. (G) Dot plot of markers first detected in MEP glia, in percent. (H) Control and *foxd3* mRNA-injected *nkx2.2a:nls-egfp;foxd3:mcherry* embryos showing a *nkx2.2a*[+]/*foxd3*[+] MEP glia (outlined arrowhead) in the PNS in a control embryo and *nkx2.2a*[+]/*foxd3*[+] MEP glia both in the CNS (arrowhead) and PNS (outlined arrowhead) in the injected embryo, at 50 hpf. (I), Mean ± SEM of additional CNS-located MEP glia indicates 0.08 ± 0.03 MEP glia in control embryos (n = 76 hemi-segments from 10 embryos), and 0.54 ± 0.07 MEP glia in *foxd3* mRNA injected embryos (n = 78 hemi-segments from 10 embryos) at 50 hpf; p<0.0001. Asterisks denote the DRG and yellow dashed lines denote the edge of the spinal cord. Scale bar, (A, F) 25 μm, (B) 0.5 mm, (E) 10 μm, and (H) 20 μm.

The online version of this article includes the following source data and figure supplement(s) for figure 3:

**Source data 1.** Source data for *Figure 3* .

**Figure supplement 1.** MEP glia do not exit the spinal cord in*foxd3*mutant larvae.

**Figure supplement 2.** Schematic of the zebrafish neural tube showing *foxd3*-dependent MEP glial (purple) delamination from the lateral floor plate (LFP), mirroring *foxd3*-dependent neural crest cell (red) migration from the dorsal neural tube.

To begin to elucidate how *foxd3* mediates MEP glial exit from the spinal cord, we investigated the role that this transcription factor plays in the development of the neural crest. Electroporation experiments in chick show that ectopic overexpression of *foxd3* in the neural tube leads to the delamination of supernumerary neural crest like-cells (*Dottori et al., 2001*). Therefore, we hypothesized that loss of *foxd3* may lead to a failure of delamination of MEP glia from the precursor domain and overexpression would promote it. To test this hypothesis, we injected *foxd3* mRNA into one-cell stage *nkx2.2a:nls-egfp;foxd3:mcherry* embryos. At 55 hpf, we observed supernumerary *nkx2.2*[+]/*foxd3*[+] MEP glia-like cells in the CNS of injected embryos (*Figure 3H,I*). When we time-lapse imaged these larvae from 50 to 72 hpf, we observed that while a single MEP glial cell exited the control spinal cord (*Video 3*), a supernumerary MEP glial-like cell generated in the lateral floor plate of *foxd3* mRNA-injected larvae eventually exited the spinal cord at MEP TZs (*Video 4*). From these experiments, we conclude that MEP glia delaminate from the lateral floor plate of the spinal cord in a *foxd3*-dependent manner, analogous to neural crest cell *foxd3*-dependent delamination and migration (*Figure 3—figure supplement 2*).

Because *foxd3* is a transcription factor, we sought to elucidate the downstream molecular mechanisms underlying MEP glial delamination and migration (*Cheung et al., 2005; Dottori et al., 2001; Hanna et al., 2002; Kos et al., 2001; Pohl and Knöchel, 2001*). The boundary between the CNS and PNS is composed of radial glial endfeet that secrete extracellular matrix components to establish a basal lamina (*Fraher et al., 2007; Lee and Song, 2013; Smith et al., 2016*). Cells, including neurons and their growth cones, secrete enzymes such as matrix metalloproteinases (MMPs) to degrade proteins of the extracellular matrix in order to migrate through it (*Hehr et al., 2005; McFarlane, 2003; Yong et al., 2001*). Therefore, we hypothesized that MEP glial exit from the

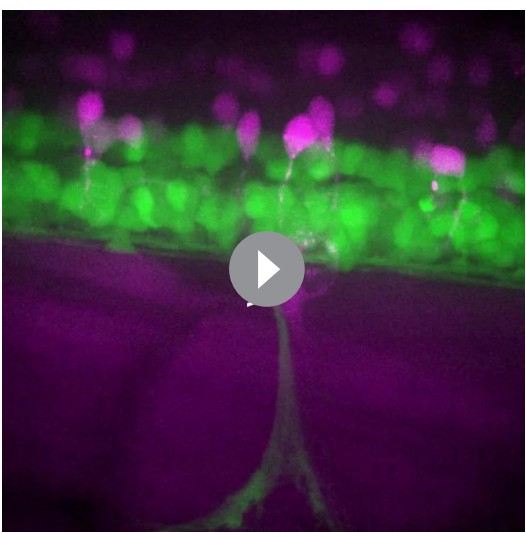

**Video 1.** *foxd3*[+] MEP glia (arrow) exit the spinal cord and migrate onto *olig2*[+] motor axons in *foxd3:mcherry; olig2:egfp;foxd3*[+/-] larvae between 48 and 72 hpf. Images were taken every 15 min and the movie runs at 10 frames per second (fps).

https://elifesciences.org/articles/64267#video1

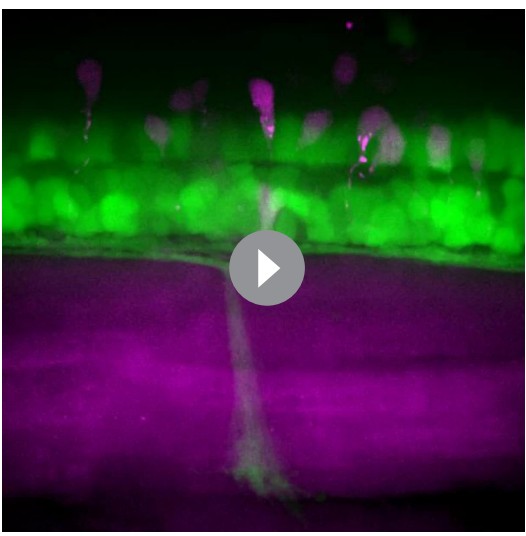

**Video 2.** *foxd3+* MEP glia are stalled in the spinal cord (arrow) in *foxd3:mcherry;olig2:egfp;foxd3-/-* larvae imaged between 48 and 72 hpf. Images were taken every 15 min and the movie runs at 10 fps.
https://elifesciences.org/articles/64267#video2

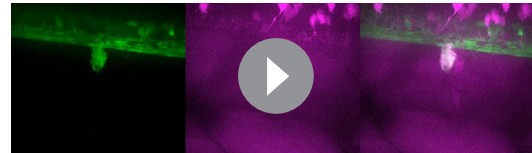

**Video 3.** A single *nkx2.2a+/foxd3+* MEP glia (arrow) has exited the spinal cord and divides in the PNS in control *foxd3:mcherry;nkx2.2a:megfp;nkx2.2a:nls-egfp* embryos. Video on the right is the merge of GFP and mCherry channels. Images were taken every 15 min from 48 to 72 hpf and the movie runs at 10 fps.
https://elifesciences.org/articles/64267#video3

CNS requires extracellular matrix degradation. To test this hypothesis, we inhibited MMPs prior to MEP glial exit. We treated *nkx2.2a:megfp; foxd3:mcherry* embryos with either 100 µM of the global MMP inhibitor GM6001 in 1% DMSO, or 1% DMSO alone, from 36 hpf on and time-lapse imaged MEP glial migration at TZs from 45 to 55 hpf and from 55 hpf to 72 hpf. At 48 hpf, we observed *nkx2.2a+/foxd3+* MEP glia delaminate from the lateral floor plate and exit the spinal cord in DMSO-treated control larvae (*Figure 4A* and *Video 5*). In contrast, in GM6001-treated larvae, we detected delamination of *nkx2.2a+/foxd3+* MEP glia from the lateral floor plate, but they stalled at the edge of the CNS/PNS boundary and rarely exited into the periphery (*Figure 4A* and *Video 6*). To determine if MEP glial exit was delayed under these conditions, we quantified the number of MEP glia present in the spinal cord or along motor nerves at 55 and 72 hpf (*Figure 4B,C*). Quantification revealed that while 90.61% MEP glia exited the spinal cord in control larvae by 55 hpf, only 8.58% migrated out of the spinal cord when MMPs were inhibited, and 53.01% were stalled in the CNS (*Figure 4B*). By 72 hpf, 100% of MEP glia were found in the PNS of control larvae, whereas 46.65% were located in the spinal cord of GM6001-treated larvae and only 31.78% were present along spinal motor nerves (*Figure 4C*). To visualize the location of the stalled MEP glia, we imaged transverse sections of control or drug-treated *foxd3:mcherry;nkx2.2a:megfp* larvae and observed *foxd3+/nkx2.2a+* MEP glia in the PNS of DMSO-treated larvae (*Figure 4D*). In contrast, we often observed *foxd3+/nkx2.2a+* MEP glia stalled in the CNS of GM6001-treated larvae at 72 hpf, in a position adjacent to the midline (*Figure 4D*). From these results, we conclude that blocking MMPs inhibits MEP glial migration out of the spinal cord and phenocopies the *foxd3* mutant phenotype.

Global inhibition of MMPs phenocopied the MEP glial phenotype we observed in *foxd3-/-* larvae and this led us to hypothesize that *foxd3* controls MEP glial exit through the regulation of MMP expression. MMPs are known to play a role in NCC migration in zebrafish and chick (*Andrieu et al., 2020*; *Leigh et al., 2013*). Therefore, we asked whether expression of MMPs was perturbed in the ventral neural tube of *foxd3* mutant embryos. We investigated the expression of the matrix metalloproteinases *mmp17b* and *adamts3* in WT embryos by in situ hybridization because they have previously been described as involved in NCC migration or were expressed in the embryonic nervous system in the vicinity of MEP TZs (*Andrieu et al., 2020*; *Asakawa et al., 2013*; *Leigh et al., 2013*). From

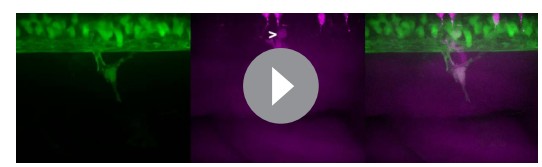

**Video 4.** An additional *nkx2.2a+/foxd3+* MEP glia (arrow) exits the spinal cord in *foxd3* mRNA-injected *foxd3:mcherry;nkx2.2a:megfp;nkx2.2a:nls-egfp* embryos. Video on the right is the merge of GFP and mCherry channels. Images were taken every 15 min from 50 to 72 hpf and the movie runs at 10 fps.
https://elifesciences.org/articles/64267#video4

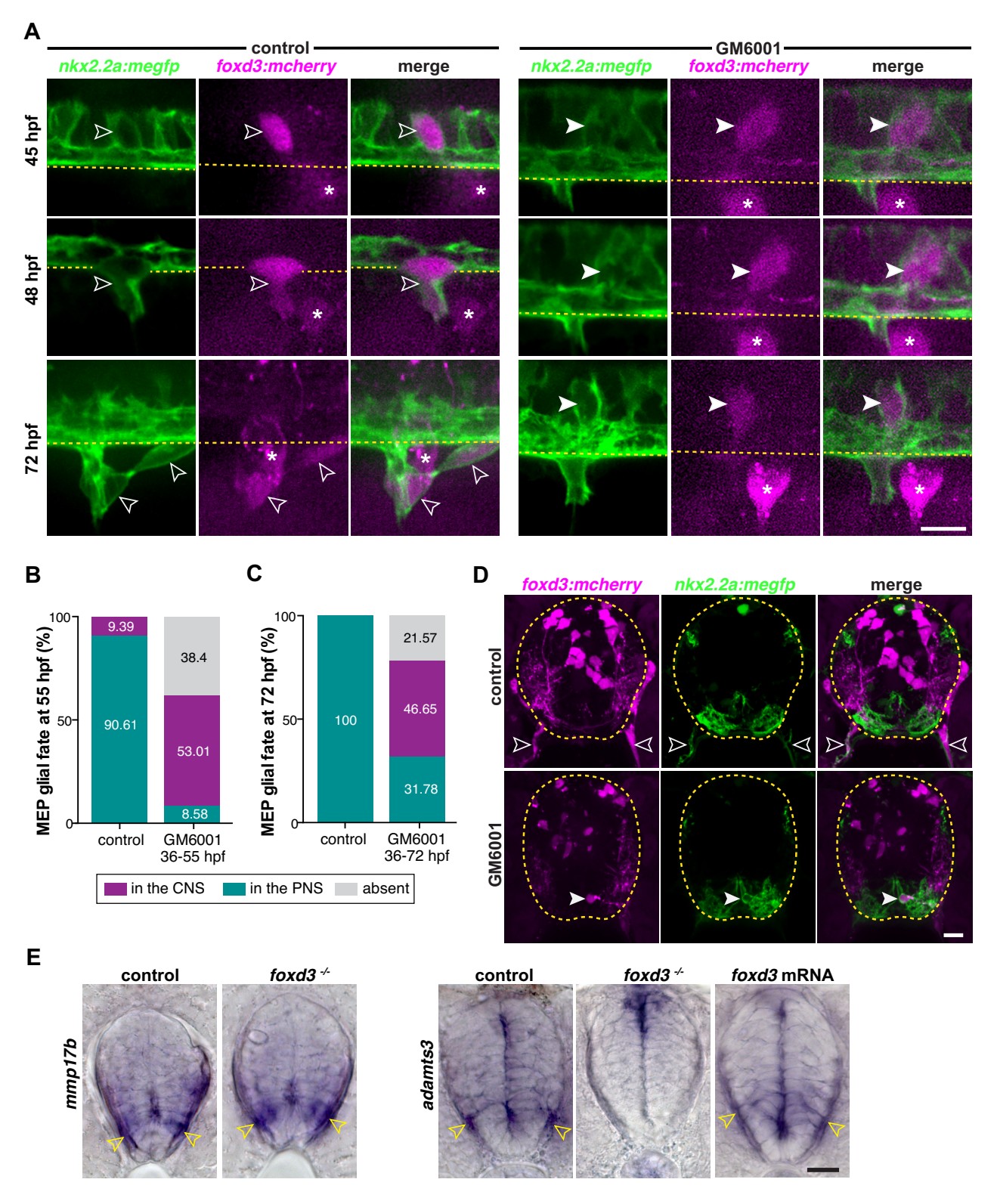

**Figure 4.** Matrix metalloproteinases participate in MEP glial exit. (**A**) Lateral view of the zebrafish trunk showing a *nkx2.2a⁺/foxd3⁺* MEP glial cell (outlined arrowhead) delaminate from the lateral floor plate and exit the spinal cord in a control embryo and *nkx2.2a⁺/foxd3⁺* MEP glia unable to exit the spinal cord (white arrowhead) in an embryo treated with 100 μM GM6001 from 36 to 48 hpf. (**B**) Percentage of MEP glia in the spinal cord, absent, or present along motor nerves at 55 hpf in DMSO control (n = 68 nerves from six embryos) and GM6001- treated embryos (n = 68 nerves from eight

*Figure 4 continued on next page*

*Figure 4 continued*

embryos). (C) Percentage of MEP glia in the spinal cord, absent, or present along motor nerves at 72 hpf in DMSO control (n = 70 nerves from seven larvae) and GM6001-treated larvae (n = 70 nerves from seven larvae). (D) Transverse sections of *foxd3:mcherry;nkx2.2a:megfp* spinal cords at 72 hpf showing *nkx2.2a*[+]*/foxd3*[+] MEP glia in the PNS (outlined arrowheads) in DMSO control and *nkx2.2a*[+]*/foxd3*[+] MEP glia in the lateral floor plate (arrowhead) in GM6001-treated larvae. Yellow dashed lines outline the edge of the spinal cord. (E) In situ hybridizations showing *mmp17b* in the spinal cord of *foxd3*[+/-] and *foxd3*[-/-] siblings and adamts3 expression in the spinal cord of foxd3[+/-] and foxd3[-/-] siblings and *foxd3* mRNA injected embryos at 48 hpf. Scale bar, (A, D, E) 10 μm.

The online version of this article includes the following source data for figure 4:

**Source data 1.** Source data for *Figure 4* .

these studies, we found that *mmp17b* and *adamts3* were expressed in the ventral neural tube at 48 hpf, prior to MEP glial exit (*Figure 4E*). We then assayed their expression in *foxd3* mutant embryos to determine if expression of these genes changed. In *foxd3*[-/-] embryos, we observed that while expression of *mmp17b* was unchanged in the spinal cord, *adamts3* expression was significantly reduced in the ventral neural tube at 48 hpf (*Figure 4E*). However, we did not observe any change in *adamts3* expression in the spinal cord of embryos overexpressing *foxd3* (*Figure 4E*). Reciprocally, overexpression of *human adamTS3* mRNA did not result in more MEP glial delamination from the precursor domain (control: mean ± SEM = 1.06 ± 0.056, n = 18 nerves from five embryos; *adamts3* mRNA: mean ± SEM = 1 ± 0.00, n = 29 nerves from nine embryos; p=0.38). Together, these experiments demonstrate the role of MMPs in MEP glial exit and identify *adamts3* as one possible candidate downstream of *foxd3* signaling in this process.

## Neural tube precursors generate one MEP glial cell progenitor per hemi-segment

In our *foxd3* mutant analysis, we always observed the presence of one MEP glial cell per somite, stalled inside the spinal cord, just medial to the MEP TZ, in a repetitive pattern along the anterior-posterior axis of the neural tube (*Figure 3A,E and F*). Additionally, our time-lapse movies consistently showed one cell exiting the spinal cord at MEP TZs and dividing in the periphery in control embryos (*Video 7*). This led us to hypothesize that radial glial precursors give rise to a single MEP glial cell progenitor per hemi-segment. To test this hypothesis, we used Zebrabow (zebrafish Brainbow), a multispectral cell labeling tool developed for cell tracing and lineage analysis (*Brockway et al., 2019*; *Pan et al., 2013*). Zebrabow consists of Cre recombinase-driven, random combinatorial expression of three spectrally distinct fluorescent proteins: cyan, red, and yellow (CFP, RFP, and YFP, respectively). The random combination of fluorescent proteins resulting from Cre recombination results in more than 30 colors and provides a way to distinguish adjacent cells for lineage analysis. Colors are inherited equally among daughter cells and remain stable throughout development (*Pan et al., 2013*). We used the broadly expressed Zebrabow *ubi:zebrabow* transgenic line that drives RFP expression ubiquitously, and we created a tissue-specific *nkx2.2a:cre* transgenic line that drives expression of Cre recombinase in *nkx2.2a*[+] cells. As a first step, to make sure we were lineage-tracing MEP glia, as perineurial glia would also be labeled with our *nkx2.2a:cre* transgenic line (*Kucenas et al., 2008b*), we performed an immunostaining against Sox10 and examined *nkx2.2a*[+]*/Sox10*[+] MEP glia present along motor axons in *nkx2.2a:cre;ubi:zebrabow* larvae (*Figure 5A*). When we looked at the color profile of 214 *nkx2.2a*[+]*/Sox10*[+] MEP glia along 60 motor nerves at 4 dpf in confocal z-stack images, we found that 96.67% of MEP glia present along a single motor nerve were the same color, and therefore,

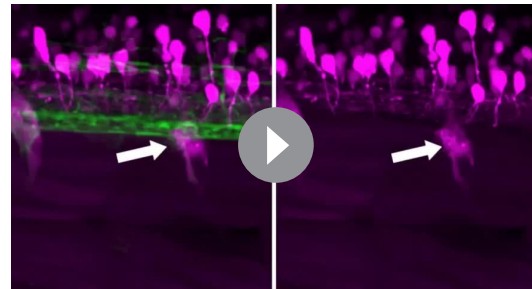

**Video 5.** One *foxd3*[+]*/nkx2.2a*[+] MEP glial progenitor delaminates from the lateral floor plate, exits the spinal cord, and divides to give rise to several MEP glia along the motor nerve in a *foxd3:mcherry;nkx2.2a:megfp* control embryo. Images were taken every 10 min from 48 to 72 hpf and the movie runs at 10 fps.
https://elifesciences.org/articles/64267#video5

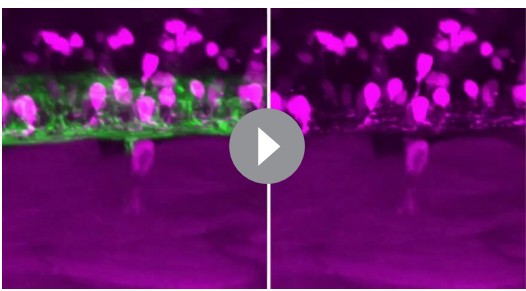

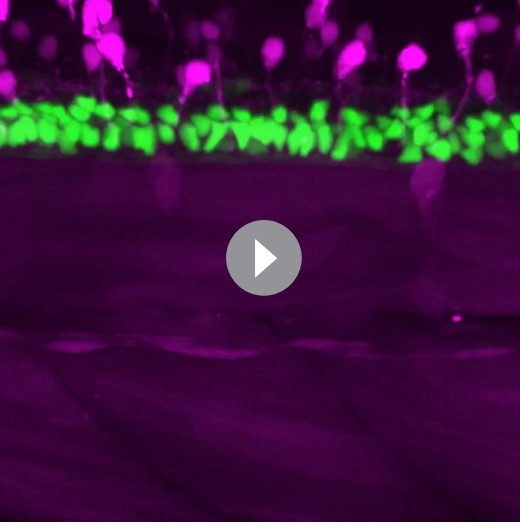

**Video 6.** One *foxd3⁺/nkx2.2a⁺* MEP glial progenitor delaminates from the lateral floor plate and stalls at the MEP in the spinal cord in a *foxd3:mcherry;nkx2.2a:megfp* embryo treated with GM6001 from 36 to 72 hpf. Images were taken every 10 min from 48 to 72 hpf and the movie runs at 10 fps.

https://elifesciences.org/articles/64267#video6

**Video 7.** One *nkx2.2a⁺/foxd3⁺* MEP glial progenitor exits the spinal cord at the MEP transition zone and undergoes cell division in the PNS to generate several MEP glia in a *nkx2.2a:nls-egfp/foxd3:mcherry* embryo. Images were taken every 15 min from 50 to 72 hpf and the movie runs at 10 fps.

https://elifesciences.org/articles/64267#video7

originated from the same *nkx2.2a⁺* precursor (*Figure 5B*).

In order to illustrate the relationship between color profile and lineage in our zebrabow paradigm in a more thorough and unbiased way, we analyzed the color profile of 14 individual MEP glia distributed along three adjacent motor nerves by measuring the fluorescence intensity in the RFP, YFP, and CFP channels separately and plotted the percentage of red, green (YFP was acquired in the green channel), and blue fluorescence (*Figure 5C*). Using a ternary plot, we observed that the color profiles of MEP glia that were present along the same motor nerve always clustered together (yellow dots cluster together, pink dots cluster together, and white dots cluster together), while fluorescent profiles of MEP glia present along distinct nerves segregated, indicating that MEP glia along the same nerve share a common progenitor, and that each hemi-segment has a single, unique MEP glial progenitor (*Figure 5C*).

To confirm that MEP glia populate motor nerves by undergoing cell division, we assessed MEP glial proliferation shortly after they exited the spinal cord. To do this, we treated *nkx2.2a:nls-egfp* embryos with EdU (5-ethynyl-2′-deoxyuridine) from 50 to 56 hpf, in order to detect DNA synthesis in proliferative cells (*Figure 5—figure supplement 1A*). To quantify proliferative MEP glia, we did an immunostaining against Sox10 and counted the percentage of *nkx2.2a⁺/Sox10⁺/EdU⁺* cells in the PNS at 72 hpf (*Figure 5—figure supplement 1B–C*). We found that 74% of the MEP glia we imaged at 72 hpf were EdU⁺ and therefore, proliferated between 50 and 56 hpf. We also quantified MEP glial cell division in our time-lapse movies between 48 and 72 hpf and observed 1 to 4 MEP glial cell divisions along motor nerve axons (*Figure 5D*), which generated up to six cells per nerve by 4 dpf (*Figure 5—figure supplement 1D*). From these studies we conclude that MEP glia populate motor nerve roots by undergoing cell division.

As an independent approach to investigating whether a single CNS-derived MEP glial cell divides several times to give rise to all the MEP glia located on a single motor nerve, we blocked cell proliferation just as MEP glia exit the spinal cord. To do this, we treated *nkx2.2a:nls-egfp;foxd3:mcherry* embryos at 44 hpf with a combination of 150 μm aphidicolin and 20 mM hydroxyurea (HUA) in order to arrest the cell cycle at approximately 48 hpf (*Lyons et al., 2005*). A previous study has demonstrated that just 4 hr of exposure to this cocktail robustly inhibits proliferation of glia in zebrafish (*Lyons et al., 2005*). When we imaged larvae at 72 hpf, we detected several MEP glia per motor nerve in DMSO-treated larvae, but only saw one MEP glial cell per hemi-segment along motor nerve roots in aphidicolin and HUA-treated larvae (*Figure 5E,F* and *Figure 5—figure supplement 1D*). This data supports our hypothesis that CNS precursors give rise to a single MEP glial progenitor cell per hemi-segment in the neural tube, and that this cell exits the CNS and acts as a progenitor that further divides to give rise to all MEP glia found on a single motor nerve.

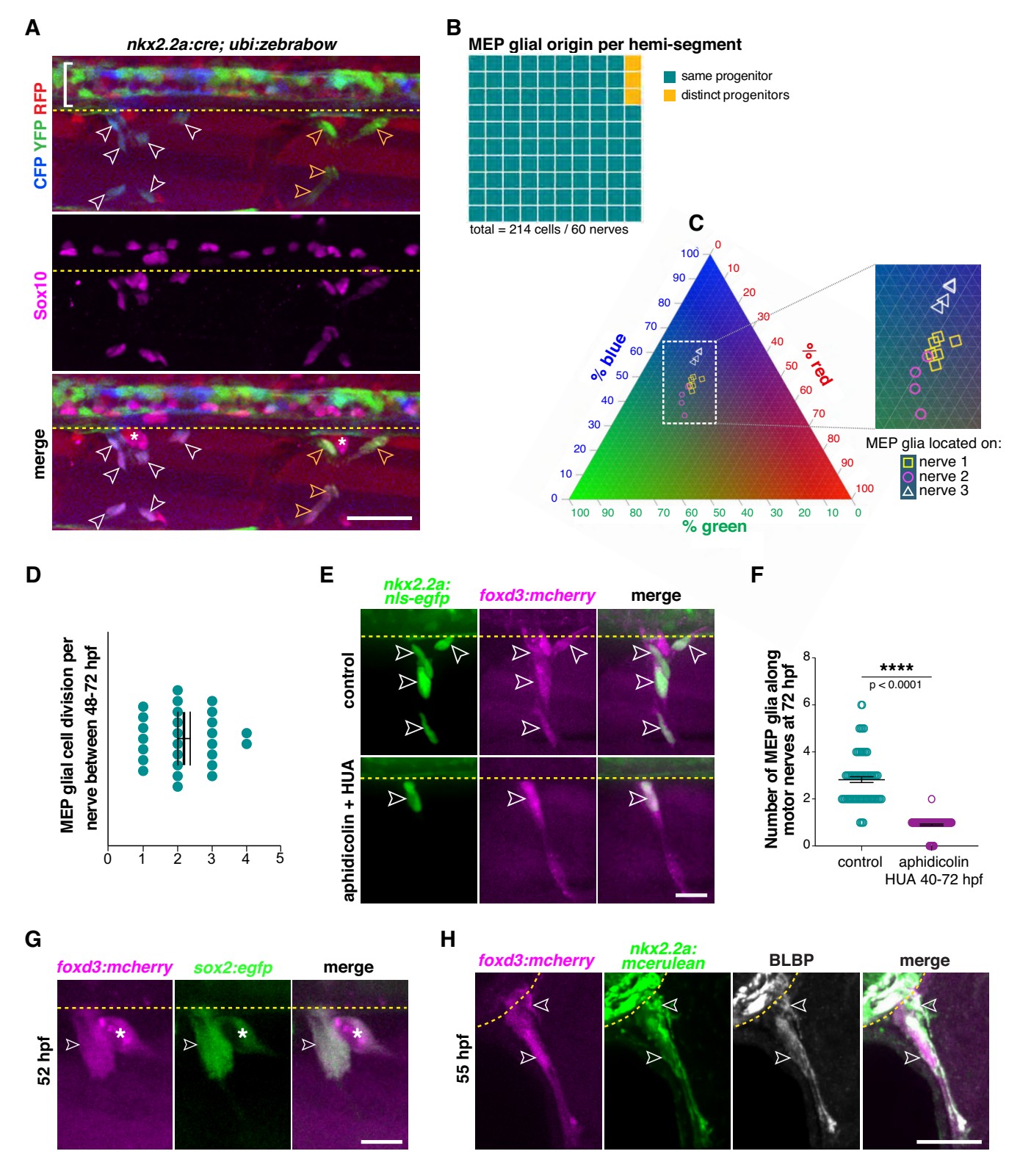

**Figure 5.** Clonal analysis reveals that neural tube precursors give rise to one MEP glial progenitor per motor exit point. (**A**) Lateral view of Sox10 immunohistochemistry in a *nkx2.2a:cre;ubi:zebrabow* larvae showing Cre recombinase driven recombination (blue and green) in *nkx2.2a*[+] cells at 4 dpf. Floor plate cells (bracket) and Sox10[+] MEP glia along two distinct nerves (white and yellow outlined arrowheads respectively) labeled. (**B**) Percentage of MEP glia present along a single motor nerve sharing a common (96.67%) vs distinct neural tube precursor (3.33%); n = 214 MEP glia and n = 60 motor

*Figure 5 continued on next page*

*Figure 5 continued*

nerves in 20 animals. (C) Ternary plot showing the intensity profile of 14 MEP glia present on three adjacent motor nerves (white and yellow symbols in C correspond to white and yellow outlined arrowheads in A) for RFP, YFP and CFP, in percent. (D) Mean ± SEM of MEP glial cell divisions per motor nerve (2.19 ± 0.18) between 48 and 72 hpf; n = 27 nerves in 12 larvae. (E) DMSO control and aphidicolin/hydroxyurea (HUA)-treated *nkx2.2a:nls-egfp; foxd3:mcherry* larvae showing *nkx2.2a⁺/ foxd3⁺* MEP glia (outlined arrowheads) along motor nerve root axons at 72 hpf. (F) Mean ± SEM of MEP glia along motor nerve roots at 72 hpf indicating 2.82 ± 0.12 MEP glia in control larvae (n = 79 hemi-segments from 10 larvae), and 0.90 ± 0.04 MEP glia in aphidicolin/HUA treated larvae (n = 89 hemi-segments from 10 larvae; p<0.0001). (G) Lateral view of a *foxd3:mcherry;sox2:egfp* larvae at 55 hpf showing *foxd3⁺/sox2⁺* MEP glia (white outlined arrowhead) and DRG (asterisk). (H) Singe z plane transverse section of a BLBP immunostaining in *foxd3:mcherry; nkx2.2a:mcerulean* larvae showing BLBP⁺/*foxd3⁺/nkx2.2a⁺* MEP glia (outlined arrowheads) outside the spinal cord along the motor nerve root at 55 hpf. Asterisks denote the DRG and yellow dashed lines denote the edge of the spinal cord. Scale bar, (A) 25 µm, (E, H) 20 µm, (G) 10 µm.

The online version of this article includes the following source data and figure supplement(s) for figure 5:

**Source data 1.** Source data for *Figure 5* .
**Figure supplement 1.** MEP glial proliferation.
**Figure supplement 1—source data 1.** Source data for *Figure 5—figure supplement 1* .

To test whether MEP glia have other progenitor characteristics, we created a *sox2:egfp* transgenic line where *sox2* regulatory sequences drive eGFP in neural progenitors. We imaged *sox2: egfp;foxd3:mcherry* embryos and larvae and detected *foxd3⁺/sox2⁺* MEP glia in the PNS at 52 hpf (*Figure 5G*) and as late as 5 dpf (*Figure 5—figure supplement 1E*), revealing that some MEP glia maintain a progenitor state. In light of these observations, we performed an immunostaining directed against another neural progenitor marker, brain lipid binding protein (BLBP), also known as *fatty acid binding protein 7* (*fabp7*), in *nkx2.2a:mcerulean;foxd3:mcherry* larvae at 55 hpf. In these larvae, we observed BLBP⁺/*nkx2.2a⁺/foxd3⁺* MEP glia along motor nerve root axons at 55 hpf (*Figure 5H*). Together, these results show that MEP glia along the same nerve come from a single progenitor and divide several times to populate spinal motor nerve root axons.

## MEP glial development requires axonal cues

Once in the periphery, MEP glia migrate along spinal motor root axons. Because they reside along the same axons as Schwann cells, and because interactions with axons are essential for Schwann cell migration, development, and function (*Fontenas et al., 2016*; *Gilmour et al., 2002*; *Lyons et al., 2005*; *Miyamoto et al., 2017*; *Perlin et al., 2011*; *Voas et al., 2007*), we hypothesized that MEP glial development also depends on axonal cues. To examine whether axonal cues are sufficient to attract MEP glia from the spinal cord to the PNS, we first took advantage of *plexinA3* mutants, also known as *sidetracked*, which lack the semaphorin receptor plexinA3 (*Palaisa and Granato, 2007*). *PlexinA3* mutants are characterized by intraspinal motor axon guidance defects that result in ectopic, supernumerary MEP TZs where motor axons ectopically exit into the periphery. We hypothesized that if axons were sufficient to attract MEP glia out of the spinal cord, *plexinA3⁻/⁻* ectopic motor axons would be associated with MEP glia. We used *sox10:eos;nbt:dsred* embryos, where *nbt* regulatory sequences label all neurons and axons (*Peri and Nüsslein-Volhard, 2008*). We then used UV light-induced photoconversion of *sox10:eos* embryos at 48 hpf to distinguish between red, photoconverted neural crest-derived Schwann cells from green, non-photoconverted MEP glia and OL lineage cells, as previously described (*Smith et al., 2014*). We then imaged photoconverted *plexina3;nbt:dsred;sox10:eos* mutant and WT siblings from 48 to 72 hpf and observed the presence of MEP glia along every motor nerve, including ectopic ones, in both mutant and WT larvae (*Figure 6A* and *Figure 6—figure supplement 1A–B*). From this data, we conclude that motor axons are sufficient to induce MEP glial migration into the periphery.

In order to confirm the role of motor axons in MEP glial migration, we sought to examine if motor axons were required for MEP glial development. To do this, we used *sox10:tagrfp;hb9:yfp-ntr* embryos, where *hb9* regulatory sequences drive expression of YFP and the bacterial enzyme nitroreductase in motor neurons (*Mathias et al., 2014*; *Zhu et al., 2019*). Nitroreductase converts the prodrug metronidazole (MTZ) into a cytotoxic drug and allows for temporally controlled, targeted-cell ablation (*Curado et al., 2008*). We treated *hb9:yfp-ntr;sox10:tagrfp* embryos with 20 mM MTZ in 2% DMSO starting from 9 hpf, prior to motor neuron specification, and assessed the presence of MEP glia in the periphery using in vivo, time-lapse imaging between 48 and 72 hpf. In these studies, we did not detect any MEP glial migration out of the spinal cord at MEP TZs where motor nerves

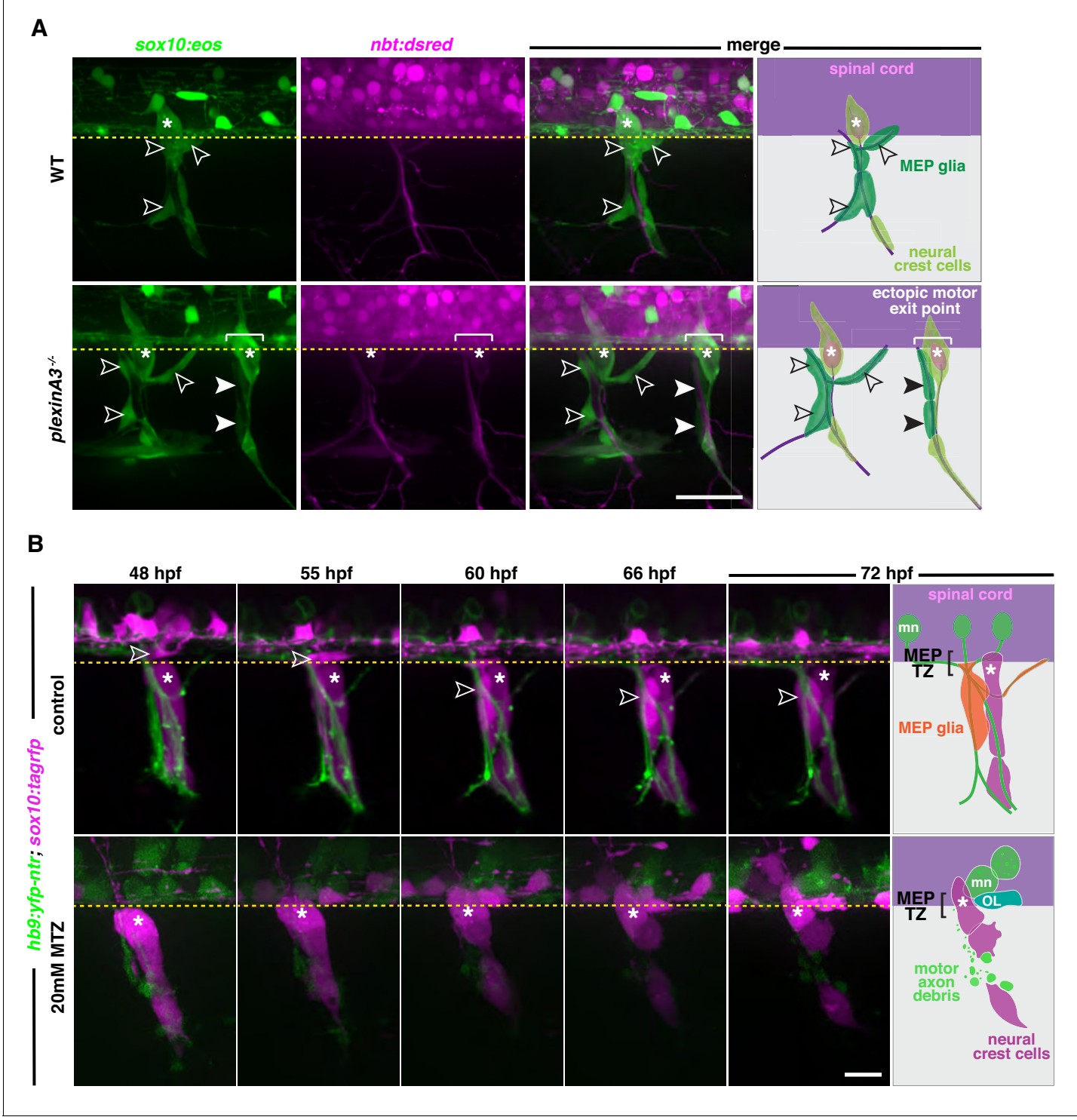

**Figure 6.** MEP glial development is axonal dependent. (A) Lateral view of *sox10:eos;nbt:dsred* WT control and *plexinA3⁻/⁻* siblings at 3 dpf showing MEP glia along both motor nerve axons (outlined arrowheads) and ectopic motor nerve axons (arrowheads). Brackets denote ectopic motor exit points. (B) Lateral views of *sox10:tagrfp;hb9:yfp-ntr* larvae from 48 to 72 hpf, treated with either 2% DMSO or 20 mM metronidazole(MTZ)/2% DMSO from 9 to 72 hpf, showing *sox10⁺* MEP glia (white outlined arrowheads) exit the spinal cord and migrate onto healthy motor axons in the control embryo. In the MTZ-treated larvae, MEP glia did not exit the spinal cord. Asterisks denote the DRG and yellow dashed lines denote the edge of the spinal cord. Scale bar, (A) 25 µm, (B) 20 µm.

The online version of this article includes the following figure supplement(s) for figure 6:

**Figure supplement 1.** Ectopic motor axons are associated with ectopic MEP glia in plexinA3 mutant larvae.

*Figure 6 continued on next page*

**Figure supplement 2.** MEP glia do not exit the spinal cord in the absence of motor axons.

were genetically ablated (*Figure 6B*). In these time-lapse movies, OPCs in the spinal cord and Schwann cells along sensory nerves both developed and migrated normally. To ensure that the absence of MEP glia in the PNS was not due to a specification failure, we examined expression of *wif1* at the conclusion of our time-lapse acquisition. At 72 hpf, we detected *wif1*⁺ MEP glia along motor roots in control larvae (*Figure 6—figure supplement 2A*). In contrast, we only observed *wif1* expression inside the spinal cord of MTZ-treated *hb9:yfp-ntr* larvae (*Figure 6—figure supplement 2A*). To better visualize MEP glia in the spinal cord of motor neuron ablated embryos, we treated *hb9:yfp-ntr;foxd3:mcherry* embryos with MTZ and looked at *foxd3*⁺ MEP glia between 50 and 72 hpf. While we saw *foxd3*⁺ peripheral MEP glia along motor axons in control larvae, we only saw spinal cord-located *foxd3*⁺ MEP glia in MTZ-treated larvae at both 50 and 72 hpf and did not see any along degenerating motor axons (*Figure 6—figure supplement 2B*). We conclude from these experiments that spinal motor axons are necessary and sufficient to drive MEP glial migration into the PNS.

### *Neuregulin 1 type III* drives MEP glial migration

Numerous studies demonstrate that axonally-derived Neuregulin1 type III is essential for many aspects of Schwann cell development, including directed migration, proliferation, and myelination along motor axons (*Heermann and Schwab, 2013*; *Lee et al., 2020*; *Lyons et al., 2005*; *Meyer et al., 1997*; *Miyamoto et al., 2017*; *Newbern and Birchmeier, 2010*; *Perlin et al., 2011*; *Taveggia et al., 2005*). We previously observed that MEP glia are absent along spinal motor nerves in *erbb3b* mutants and *erbb3b* is expressed all along the motor nerve including at motor roots (*Langworthy and Appel, 2012*; *Smith et al., 2014*). Therefore, we chose to examine the role of *neuregulin 1 type III* (*nrg1*) in MEP glial migration. We first investigated the presence of MEP glia in *nrg1* zebrafish mutants (*Perlin et al., 2011*) by using photoconversion to differentially label MEP glia and Schwann cells and live imaging in *sox10:eos* embryos (*McGraw et al., 2012*). In *nrg1*⁺/⁺;*sox10:eos* larvae, we observed red, photoconverted DRG and Schwann cells, and green, non-photoconverted MEP glia along spinal motor nerve root axons (*Figure 7A*). In contrast, in *nrg1*⁻/⁻;*sox10:eos* larvae, MEP glia and Schwann cells were absent from motor nerve roots at 72 hpf (*Figure 7A*), consistent with a recent study showing a reduction in *sox10*⁺ cell numbers along motor nerves in *nrg1* morpholino-mediated knockdown zebrafish larvae (*Lee et al., 2020*). Consistent with other previous work, we also did not detect the presence of neural-crest-derived DRG in *nrg1* mutants (*Honjo et al., 2008*). We did, however, observe the presence of *sox10*⁺ processes projecting through the MEP TZ along motor nerve root axons in *nrg1*⁻/⁻;*sox10:eos* larvae (*Figure 7A*). To identify the origin of these processes, we segmented the confocal images to identify individual cells and pseudo-colored in white the cell that was the closest to the MEP TZ (*Figure 7A*). By analyzing its morphology, we observed that in WT larvae, the cell that is the most proximal to the MEP TZ is a MEP glial cell, located in the PNS (*Figure 7A*). In *nrg1* mutant larvae, the only cell that is present at the MEP TZ is located in the CNS, has the morphology of an OPC, and projects a process through the MEP (*Figure 7A*). To confirm that the projections we observed from within the CNS were OPC processes, we did an in situ hybridization for the oligodendrocyte myelin marker *proteolipid protein 1a* (*plp1a*), that is not expressed by peripheral myelinating glia in zebrafish (*Brösamle and Halpern, 2002*; *Kucenas et al., 2009*; *Smith et al., 2014*). We observed that while *plp1a* was expressed in the white matter of the spinal cord in WT larvae, it was also found in the PNS in *nrg1*⁻/⁻ larvae (*Figure 7—figure supplement 1A*).

We then confirmed the absence of MEP glia in *nrg1*⁻/⁻ larvae by performing an in situ hybridization for *wif1* at 3 dpf and observed *wif1*⁺ MEP glia at the MEP TZ in WT larvae (*Figure 7B* and *Figure 7—figure supplement 1B*). In contrast, we did not detect any *wif1*⁺ MEP glia along spinal motor roots in *nrg1*⁻/⁻ larvae (*Figure 7B*). However, we found *wif1*⁺ cells further distally along the motor nerve in the periphery (*Figure 7B* and *Figure 7—figure supplement 1B*), that we never observed in WT siblings and consistent with previous work showing *wif1*⁺ cells along the lateral line nerve in *nrg1* mutants (*Lush and Piotrowski, 2014*). To confirm that *nrg1* functions in MEP glia through

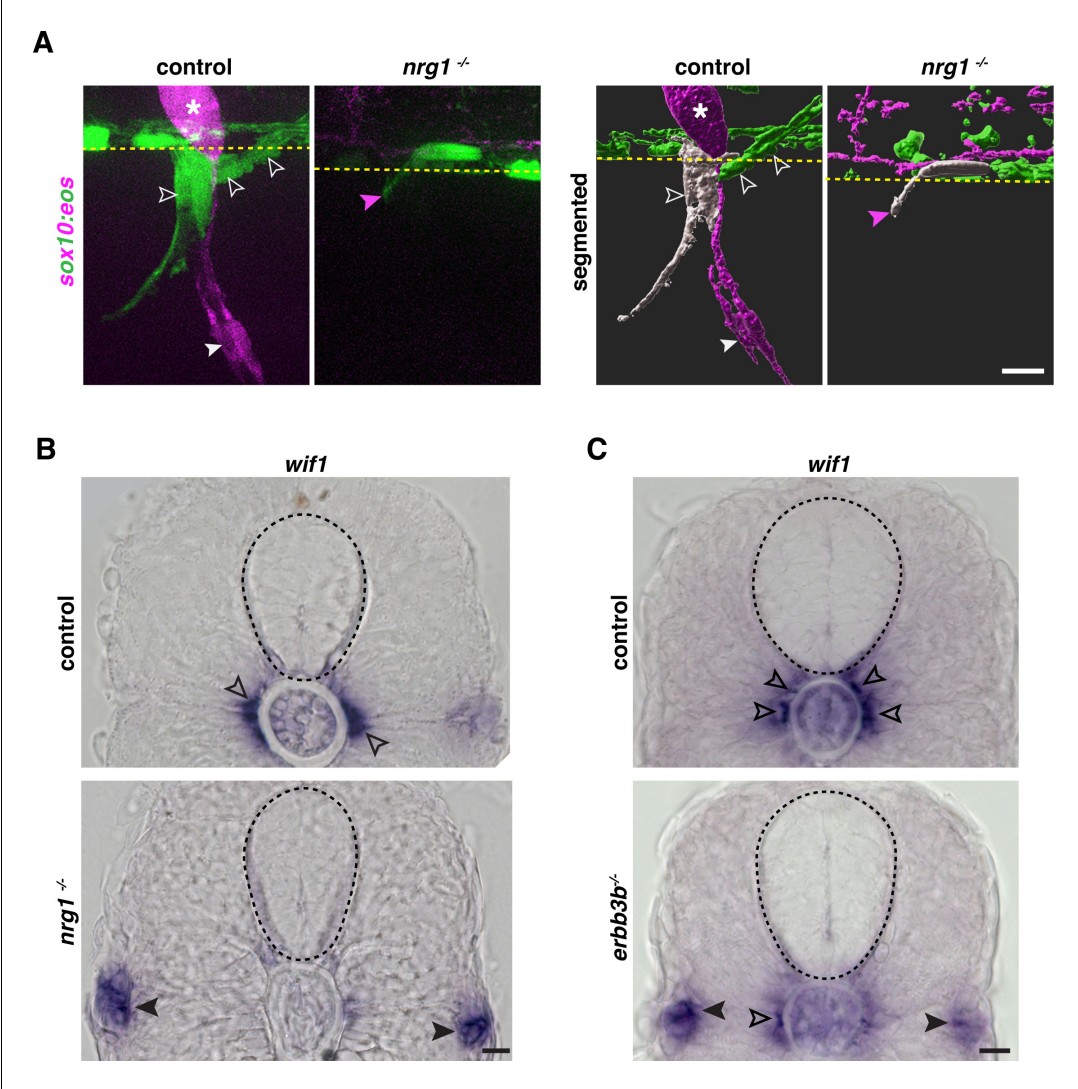

**Figure 7.** Neuregulin 1 type III drives MEP glial directed migration. (**A**) Motor exit point of *sox10:eos* WT and *nrg1* <sup>-/-</sup> siblings photoconverted at 48 hpf and imaged at 3 dpf showing the presence of MEP glia (outlined arrowheads) and Schwann cells (SCs) (arrowhead) in a control larva and the presence of an oligodendrocyte membrane extension (pink arrowhead) in a *nrg1* <sup>-/-</sup> larva that lacks MEP glia and SCs. Yellow dashed lines denote the edge of the spinal cord. (**B**) In situ hybridization showing *wif1*<sup>+</sup> MEP glia along motor nerve root axons in a WT control larva (outlined arrowheads, n = 24 larvae) and the absence of *wif1*<sup>+</sup> MEP glia along motor nerve root axons in a *nrg1* mutant larva at 3 dpf (n = 16 larvae). Arrowheads indicate the presence of *wif1*<sup>+</sup> cells near the lateral line nerve in a *nrg1* mutant larva at 3 dpf. (**C**) In situ hybridization showing *wif1*<sup>+</sup> MEP glia along motor nerve root axons in a WT control larva (outlined arrowheads, n = 20 larvae) and the absence of *wif1*<sup>+</sup> MEP glia along motor nerve root axons in an *erbb3b* mutant larva at 3 dpf (n = 14 larvae). Arrowheads indicate the presence of *wif1*<sup>+</sup> cells near the lateral line nerve in an *erbb3b* mutant larva at 3 dpf. Scale bar (**A–C**), 10 μm.

The online version of this article includes the following figure supplement(s) for figure 7:

**Figure supplement 1.** MEP glia are absent along the motor root in *nrg1* mutant larvae.

*erbb3b* receptors like in Schwann cells, we also assayed *wif1* expression in *erbb3b* mutants. In 72 hpf *erbb3b* larvae, *wif1*<sup>+</sup> cells were not exclusively found along motor root axons as in wild-type larvae (*Figure 7C*), but instead were found more distally in the trunk near the lateral line nerve (*Figure 7C*), similar to the *nrg1* mutant phenotype (*Figure 7B* and *Lush and Piotrowski, 2014*). Taken together, we conclude that *nrg1* expressed on motor axons drives MEP glial migration to the motor root, and in its absence, MEP glia exit the spinal cord, but fail to associate with motor root axons and migrate away.

To investigate whether axonally derived *nrg1* is sufficient to induce MEP glial directed migration, we genetically overexpressed human Neuregulin 1 type III (hNrg1) in most neurons, including spinal cord neurons (*Korzh et al., 1998*; *Liao et al., 1999*; *Mueller and Wullimann, 2002*), using *neuroD: gal4;UAS:hNrg1* embryos, where *neuroD1* regulatory sequences drive expression of the transcription factor Gal4 (*Fontenas et al., 2019*) and UAS enhancer sequences drive expression of human Neuregulin 1 type III (*Perlin et al., 2011*). In these embryos, we detected *krox20*⁺ Schwann cells in the white matter of the spinal cord (*Figure 8—figure supplement 1*) as previously described (*Perlin et al., 2011*). To determine if MEP glial migration was also affected by overexpression of hNrg1, we first performed an in situ hybridization for *wif1* to visualize MEP glia at 3 dpf in *neuroD: gal4;UAS:hNrg1* larvae. In *neuroD:gal4* control larvae, we observed *wif1*⁺ MEP glia exclusively in the periphery along motor root axons (*Figure 8A*). In contrast, we detected *wif1*⁺ MEP glia both in the PNS and in the ventral spinal cord in *neuroD:gal4;UAS:hNrg1* larvae (*Figure 8A*). Because our cell lineage studies showed that just one MEP glial cell is specified in the neural tube at each MEP TZ, we sought to determine whether these centrally-located MEP glia migrated into the spinal cord from the PNS, or conversely, if they were ectopically specified CNS MEP glia-like cells. To do this, we used time-lapse imaging in either *nkx2.2a:nls-egfp* or *sox10:nls-eos* larvae to visualize MEP glia in the PNS. In both *neuroD:gal4;UAS:hNrg1* and *neuroD:gal4* control larvae, we never saw any *nkx2.2a*⁺ or *sox10*⁺ peripheral MEP glia migrate into the spinal cord between 48 and 72 hpf. However, when we imaged *nkx2.2a:nls-egfp;foxd3:mcherry* larvae, which label MEP glia more specifically, we detected several *nkx2.2a*⁺/*foxd3*⁺ MEP glia in the ventrolateral spinal cord of *neuroD:gal4; UAS:hNrg1;nkx2.2a:nls-egfp;foxd3:mcherry* larvae at 3 dpf (*Figure 8B,C*), something that we rarely observed in *neuroD:gal4;nkx2.2a:nls-egfp;foxd3:mcherry* control siblings (*Figure 8B,C*).

Because Schwann cell proliferation is driven by neuregulin/Erbb signaling (*Perlin et al., 2011*; *Taveggia et al., 2005*), we hypothesized that these centrally located MEP glia originate from increased neural tube *nkx2.2a*⁺ radial glial precursor divisions. To test this hypothesis, we assessed *nkx2.2a*⁺ cell proliferation in the neural tube in *neuroD:gal4;UAS:hNrg1;nkx2.2a:nls-egfp* and *neuroD:gal4;nkx2.2a:nls-egfp* embryos by immunolabeling Phospho-HistoneH3 (PH3)⁺ mitotic cells at 48 hpf (*Figure 8D,E*). Quantification of ventral spinal cord PH3⁺/*nkx2.2a*⁺ cells revealed an increase in proliferation in *neuroD:gal4;UAS:hNrg1* larvae compared to their *neuroD:gal4* control siblings (*Figure 8D,E*). To test whether *nrg1* also drives MEP glial proliferation along motor nerves, we treated *nkx2.2a:nls-egfp* embryos with EdU from 50 to 56 hpf and performed a Sox10 immunostaining at 72 hpf to distinguish MEP glia from perineurial glia (*Figure 8—figure supplement 2A*), as described in *Figure 5—figure supplement 1*. At 72 hpf, we observed an increase in the number of EdU⁺/*nkx2.2.a*⁺/Sox10⁺ MEP glia (*Figure 8—figure supplement 2B*), as well as an increase in the total number of MEP glia present along motor nerves (*Figure 8—figure supplement 2C*). Together, these results suggest that *nrg1* regulates both MEP glial progenitors and MEP glial proliferation.

To confirm that ventral spinal cord *nkx2.2a*⁺ radial glia have the potential to generate MEP glia in post-embryonic development, we performed an immunostaining directed against the progenitor marker Sox2 in *nkx2.2a:megfp* larvae (*Figure 8F*). At 4 dpf, we detected the presence of Sox2⁺/ *nkx2.2a*⁺ radial glial progenitors in WT larvae, which is the latest stage we have assessed (*Figure 8F*). This suggests that *nkx2.2a*⁺ radial glial precursors are still present in the spinal cord in post-embryonic development and *neuregulin1* can control their proliferation.

## Discussion

During nervous system development, neural progenitors orchestrate the formation of the brain, spinal cord, and peripheral nerves by generating neurons and glia. Although components of the CNS and PNS are traditionally thought to segregate, growing numbers of studies show the selective and bidirectional permeability of nervous system TZs (*Clark et al., 2014*; *Fontenas et al., 2019*; *Garcia-Diaz et al., 2019*; *Green et al., 2019*; *Kucenas et al., 2008b*; *Smith et al., 2016*; *Smith et al., 2014*; *Zhu et al., 2019*). Development of myelinating cells such as oligodendrocytes and Schwann cells has extensively been studied, but development of MEP glia, a novel population of CNS-derived, peripheral glia that freely migrate across TZs, remains poorly understood. In this study, we demonstrate that MEP glia originate from *nkx2.2a*⁺ lateral floor plate radial glia and use NCC-associated mechanisms to migrate out of the spinal cord onto motor nerves, where they develop with mechanisms similar to that of Schwann cells.

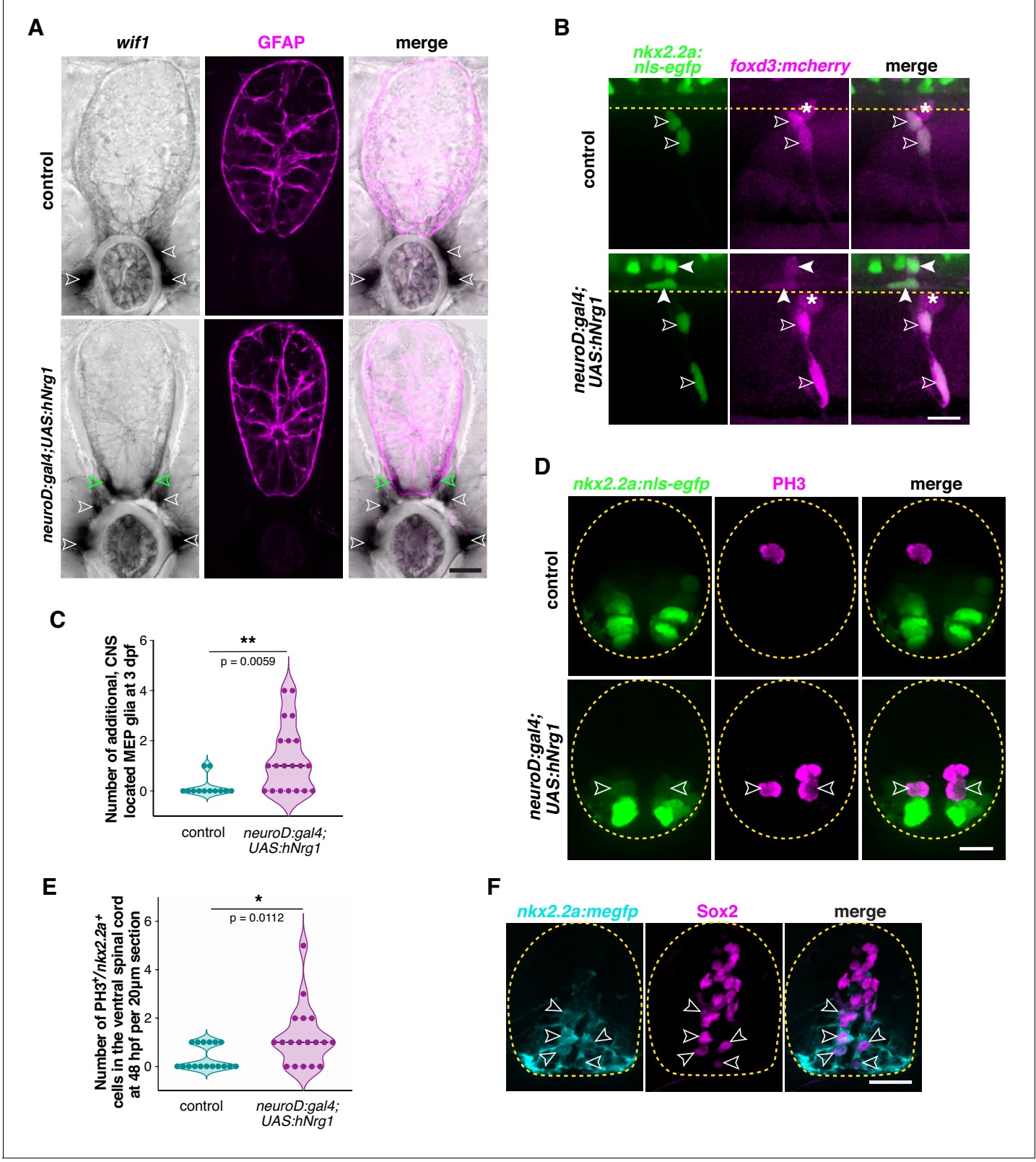

**Figure 8.** Neuregulin 1 type III drives MEP glial development. (**A**) In situ hybridization showing *wif1+* MEP glia (white outlined arrowheads) along motor nerve roots in WT larvae (n = 30 larvae) at 3 dpf and both inside (green outlined arrowheads) and outside (white outlined arrowheads) the spinal cord in *neuroD:gal4;UAS:hNrg1* larvae (n = 30 larvae). Zrf1 immunostaining shows GFAP+ radial glia and denotes the outline of the spinal cord. (**B**) Lateral view of 3 dpf *nkx2.2a:nls-egfp;foxd3:mcherry* control and *neuroD:gal4;UAS:hNrg1* larvae showing *nkx2.2a+/foxd3+* MEP glia (outlined arrowheads) along

*Figure 8 continued on next page*

*Figure 8 continued*

motor nerve root axons. MEP glia are found in both the PNS (outlined arrowheads) and in the ventral spinal cord (arrowheads) in *neuroD:gal4;UAS: hNrg1* larvae. Asterisks denote the DRG. (C) Mean ± SEM of additional, centrally located MEP glia per hemi-segment in control (0.17 ± 0.11; n = 12 larvae) and *neuroD:gal4;UAS:hNrg1* larvae (1.3 ± 0.3; n = 20 larvae) at 3 dpf. (D) PH3 immunostaining on spinal cord transverse sections in *neuroD:gal4; nkx2.2a:nls-egfp* control larvae and *neuroD:gal4;UAS:hNrg1;nkx2.2a:nls-egfp* larvae showing PH3$^+$/*nkx2.2a*$^+$ proliferating radial glia (outline arrowheads). (E) Mean ± SEM of PH3$^+$/*nkx2.2a*$^+$ cells in the ventral spinal cord per 20 µm section in *neuroD:gal4* (0.35 ± 0.12; n = 17 sections from four embryos) and *neuroD:gal4;UAS:hNrg1* larvae (1.22 ± 0.3; n = 18 sections from four embryos). (F) Transverse section of a *nkx2.2a:megfp* larva at 4 dpf showing Sox2$^+$/*nkx2.2a*$^+$ radial glial precursors. Yellow dashed lines denote the edge of the spinal cord. Scale bar (A,B,D,F), 10 µm.

The online version of this article includes the following source data and figure supplement(s) for figure 8:

**Source data 1.** Source data for *Figure 8* .

**Figure supplement 1.** In situ hybridization showing the absence of Schwann cells in the spinal cord of a control sibling (n = 20 larvae), and the presence of *krox20*$^+$ Schwann cells (white outlined arrowheads) in the ventral and dorsal spinal cord in a *neuroD:gal4;UAS:hNrg1* larva at 4 dpf (n = 20 larvae).

**Figure supplement 2.** EdU proliferation assay.

**Figure supplement 2—source data 1.** Source data for *Figure 8—figure supplement 2* .

## MEP glia originate from lateral floor plate radial glia and require *foxd3* to delaminate and exit the neural tube

The origin of the main central and peripheral myelinating glia has been known for decades. *Olig2*$^+$ pMN precursors give rise to OL lineage cells and NCCs give rise to Schwann cells (*Jessen and Mirsky, 1999*; *Park et al., 2002*; *Zhou et al., 2001*). Our observations reveal that MEP glia originate from *nkx2.2a*$^+$ lateral floor plate radial glial precursors that lie just ventral to the *olig2* pMN domain of the neural tube and identify *nkx2.2a* as a new marker for MEP glia. Our study depicts a more detailed picture of these hybrid, centrally derived, peripheral glia that express a unique set of both CNS and PNS glial markers, including *nkx2.2a, olig2, sox10, foxd3,* the boundary cap cell marker *wif1,* and make *mbp*$^+$ myelin sheaths along spinal motor nerve root axons (*Coulpier et al., 2009*; *Feng et al., 1994*; *Hochgreb-Hägele and Bronner, 2013*; *Johnson et al., 2016*; *Kucenas et al., 2008a*; *Montero-Balaguer et al., 2006*; *Qi et al., 2001*; *Smith et al., 2014*; *Soula et al., 2001*; *Zhou et al., 2001*).

*Olig2* and *nkx2.2a* are two well-known identity markers of cells born in the ventral neural tube and their expression by MEP glia was not unexpected (*Briscoe et al., 1999*; *Park, 2004*; *Park et al., 2002*; *Qi et al., 2001*; *Rowitch, 2004*; *Zhou et al., 2001*). However, *foxd3* is known as an early neural crest marker and its function in early steps of NCC development is conserved across all vertebrate species (*Fairchild et al., 2014*; *Hochgreb-Hägele and Bronner, 2013*; *Lister et al., 2006*; *Montero-Balaguer et al., 2006*; *Sasai et al., 2001*; *Teng et al., 2008*). Therefore, both its expression and function in neural tube-derived glia are thought-provoking. Studies in various vertebrate species demonstrate that *foxd3* is not necessary for neural crest specification, but is required for NCC migration (*Fairchild et al., 2014*; *Lister et al., 2006*; *Pohl and Knöchel, 2001*; *Wang et al., 2011*). Similarly, our loss-of-function study shows that *foxd3* is also not essential for MEP glial specification, but rather, is required for their delamination and exit from the neural tube and that the latter is at least partially controlled by MMPs. Interestingly, electroporation of tagged-Foxd3 in the chick neural tube leads to the delamination of Foxd3$^+$ cells from the neural tube (*Dottori et al., 2001*). When we tested whether overexpressing *foxd3* in zebrafish was sufficient to induce ectopic delamination and/or exit of lateral floor plate cells, we found an increase in the number of MEP glia that delaminate and exit the CNS.

## MEP glial development depends on axonal Neuregulin 1 type III

Our time-lapse movies and clonal analysis reveal that *nkx2.2a*$^+$ radial glia generate one MEP glia per hemi-segment, which delaminates from the *nkx2.2a* domain and exits the spinal cord through a MEP TZ. Following migration onto peripheral nerves, CNS-derived MEP glia divide and give rise to two to six cells per motor nerve by 4 dpf. We show that MEP glia are proliferative and that blocking the cell cycle when MEP glia exit the CNS results in a unique MEP glia per hemi-segment in the PNS. Consistent with this observation, our confocal imaging shows that MEP glia express BLBP and *sox2*, two markers of neural progenitors, that are important for self-renewal (*Mercurio et al., 2019*; *Owada et al., 1996*; *Sharifi et al., 2013*).

The signals that instruct MEP glia to delaminate from the ventral neural tube to migrate through MEP TZs and onto peripheral motor axons have never been explored. Our work shows that in addition to *foxd3*, MEP glia require motor axons to exit the spinal cord and are not found in the PNS in larvae where MEP TZs exist, but motor axons were genetically ablated. Conversely, ectopic motor exit points in the *sidetracked* mutants are always associated with MEP glia. We identified *neuregulin 1 type III* (*nrg1*) as one of the axonal cues that drive MEP glial migration. The axonal ligand *nrg1* and its receptor *erbb3*, expressed in SCs, are largely known to play a major role in Schwann cell myelination but also in their migration (*Lyons et al., 2005*; *Miyamoto et al., 2017*; *Newborn and Birchmeier, 2010*; *Perlin et al., 2011*). Therefore, we chose to examine the role of *nrg1* in MEP glial interaction with motor axons and MEP glial migration. Using time-lapse imaging and in situ hybridization, we observed that in *nrg1* and *erbb3b* mutant larvae, after exiting the spinal cord, MEP glia do not associate with motor root axons but instead, keep migrating ventrally toward neuro-muscular junctions. Our analysis demonstrates that MEP glia use at least one of the traditional mechanisms that drive Schwann cell development in order to migrate onto motor nerves. Previous work showed that the Neuregulin receptor *errb3b* is expressed at MEPs and that MEP glia are disrupted in *erbb3* mutants, as they are not present along motor nerve root axons in these mutants (*Langworthy and Appel, 2012*; *Morris et al., 2017*; *Smith et al., 2014*). Reciprocally, we asked whether overexpressing Neuregulin 1 type III in CNS neurons was sufficient to divert MEP glia to the spinal cord. Surprisingly, we found that unlike Schwann cells (*Perlin et al., 2011*), Nrg1 overexpression is not sufficient to reroute MEP glial migration from the PNS to the CNS, but instead, results in the specification of additional MEP glia from spinal cord *nkx2.2a*[+] radial glial precursors at post-embryonic stages (see *Figure 8C,D*). These findings raise an intriguing question: how similar are these post-embryonic CNS-located MEP glia to the CNS-born Schwann cells found in spinal cord demyelinating lesions (*Zawadzka et al., 2010*)? Future studies will be needed to determine whether Nrg1-induced CNS located MEP glia differentiate and myelinate spinal cord axons under both physiological and demyelinating conditions.

## Materials and methods

### Key resources table

| Reagent type (species) or resource | Designation | Source or reference | Identifiers | Additional information |
|---|---|---|---|---|
| Strain (*Danio rerio*) | AB* | ZIRC | RRID:ZFIN_ZDB-GENO-960809-7 | |
| Genetic reagent (*Danio rerio*) | Tg(nkx2.2a(3.5):Cre; cmlc2:eGFP)[uva42] | This paper | | |
| Genetic reagent (*Danio rerio*) | Tg(nkx2.2a(3.5): mCerulean3)[uva41] | This paper | | |
| Genetic reagent (*Danio rerio*) | Tg(nkx2.2a(3.5):nls-eGFP)[uva1] | This paper | | |
| Genetic reagent (*Danio rerio*) | Tg(mbp: eGFP-CAAX)[ue2] | *Almeida et al., 2011* | RRID:ZFIN_ZDB-ALT-120103-2 | |
| Genetic reagent (*Danio rerio*) | Gt(foxd3: mcherry)[ct110R] | *Hochgreb-Hägele and Bronner, 2013* | RRID:ZFIN_ZDB-ALT-130314-2 | |
| Genetic reagent (*Danio rerio*) | Tg(XlTubb: DsRed)[zf148] | *Peri and Nüsslein-Volhard, 2008* | RRID:ZFIN_ZDB-ALT-081027-2 | |
| Genetic reagent (*Danio rerio*) | Tg(neuroD1:Gal4; cmlc2:eGFP)[uva22] | *Fontenas et al., 2019* | RRID:ZFIN_ZDB-ALT-191209-8 | |
| Genetic reagent (*Danio rerio*) | Tg(nkx2.2a(3.5): nls-mCherry)[uva2] | *Zhu et al., 2019* | RRID:ZFIN_ZDB-ALT-200513-2 | |
| Genetic reagent (*Danio rerio*) | Tg(olig2:eGFP)[vu12] | *Shin et al., 2003* | RRID:ZFIN_ZDB-ALT-041129-8 | |

*Continued on next page*

*Continued*

| Reagent type (species) or resource | Designation | Source or reference | Identifiers | Additional information |
|---|---|---|---|---|
| Genetic reagent (*Danio rerio*) | *Tg(olig2:DsRed2)*[vu19] | *Shin et al., 2003* | RRID:ZFIN_ZDB-FISH-150901–8168 | |
| Genetic reagent (*Danio rerio*) | *Tg(sox10(4.9):Eos)*[w9] | *McGraw et al., 2012* | RRID:ZFIN_ZDB-ALT-110721-1 | |
| Genetic reagent (*Danio rerio*) | *Tg(sox10(4.9):nls-Eos)*[w18] | *McGraw et al., 2012* | RRID:ZFIN_ZDB-ALT-110721-2 | |
| Genetic reagent (*Danio rerio*) | *Tg(sox10(4.9):TagRFP)*[uva5] | *Zhu et al., 2019* | RRID:ZFIN_ZDB-ALT-200513–7 | |
| Genetic reagent (*Danio rerio*) | *Tg(sox10(7.2):mRFP)*[vu234] | *Kucenas et al., 2008b* | RRID:ZFIN_ZDB-ALT-080321–3 | |
| Genetic reagent (*Danio rerio*) | *Tg(UAS:hNrg1 type III)*[st85] | *Perlin et al., 2011* | RRID:ZFIN_ZDB-ALT-120221–8 | |
| Genetic reagent (*Danio rerio*) | *Tg(ubi:Zebrabow-M)*[a131] | *Pan et al., 2013* | RRID:ZFIN_ZDB-ALT-130816–2 | |
| Genetic reagent (*Danio rerio*) | *Tg(2xNRSE-2xMnx1-Mmu.Fos:KalTA4,5xUAS-ADV.E1b:GAP-YFP-2A-Eco.NfsBT41Q/N71S/F124T)*[lmc008] | *Mathias et al., 2014* | RRID:ZFIN_ZDB-ALT-151021–5 | |
| Genetic reagent (*Danio rerio*) | *Tg(sox2(2.9):eGFP)*[uva55] | This paper | | |
| Genetic reagent (*Danio rerio*) | *neuregulin1*[z26] | *Perlin et al., 2011* | RRID:ZFIN_ZDB-ALT-120308–1 | |
| Genetic reagent (*Danio rerio*) | *plexinA3* [p13umal] | *Palaisa and Granato, 2007* | RRID:ZFIN_ZDB-ALT-071126–1 | |
| Genetic reagent (*Danio rerio*) | *erbb3b*[st48] | *Lyons et al., 2005* | RRID:ZFIN_ZDB-ALT-050512–6 | |
| Recombinant DNA reagent | p5E-nkx2.2a(−3.5) | *Pauls et al., 2007* | N/A | |
| Recombinant DNA reagent | pME-mcerulean3 | *Zhu et al., 2019* | N/A | |
| Recombinant DNA reagent | p5E-sox2(−2.9) | This paper | | |
| Recombinant DNA reagent | pME-nls-eGFP | *Kwan et al., 2007* | N/A | |
| Recombinant DNA reagent | p3E-polyA | *Kwan et al., 2007* | N/A | |
| Recombinant DNA reagent | pDestTol2CG2 | *Kwan et al., 2007* | N/A | |
| Recombinant DNA reagent | pME-cre | This paper | | |
| Recombinant DNA reagent | Human adamts3 | Genomics-online | Cat. #ABIN3996515 | |
| Commercial assay or kit | pENTR 5′-TOPO cloning kit | Invitrogen | Cat. #K59120 | |
| Commercial assay or kit | LR clonase II plus | Invitrogen | Cat. #12538–120 | |
| Commercial assay or kit | Click-it EdU Cell proliferation kit for imaging. Alexa Fluor 647 dye | Invitrogen | Cat. #C11340 | |
| Commercial assay or kit | mMESSAGE mMACHINE sp6 transcription kit | Fisher | Cat. #AM1340 | |

*Continued on next page*

*Continued*

| Reagent type (species) or resource | Designation | Source or reference | Identifiers | Additional information |
|---|---|---|---|---|
| Chemical compound, drug | DAPI fluoromount-G | Southern Biotech | Cat. #0100–20 | |
| Chemical compound, drug | DIG RNA labeling mix | Roche | Cat. #11277073910 | |
| Chemical compound, drug | metronidazole | Sigma | Cat#M1547; CAS#443-48-1 | 20 mM |
| Chemical compound, drug | GM6001 | Enzo Life Sciences | Cat. #BML-EI300-0001; CAS# 142880-36-2 | 100 µM |
| Chemical compound, drug | aphidicolin | Sigma | Cat. #A0781; CAS# 38966-21-1 | 150 µM |
| Chemical compound, drug | hydroxyurea | Sigma | Cat. #H8627; CAS#127-07-1 | 20 mM |
| Antibody | Mouse anti-GFAP | ZIRC | Cat. #Zrf1 ; RRID:AB_10013806 | 1 :1000 |
| Antibody | Anti-digoxigenin-AP, Fab fragments from sheep | Sigma | Cat#11093274910; RRID:AB_514497 | 1:5000 |
| Antibody | Rabbit anti-sox10 | *Binari et al., 2013* | N/A | 1 :5000 |
| Antibody | chicken anti-GFP | Abcam | Cat. #ab13970; RRID:AB_300798 | 1 :500 |
| Antibody | Alexa Fluor 488 goat anti-chicken | ThermoFisher | Cat. #A-11039; RRID:AB_2534096 | 1:1000 |
| Antibody | Alexa Fluor 647 goat anti-rabbit IgG(H+L) | ThermoFisher | Cat. #A-21244; RRID:AB_2535812 | 1:1000 |
| Antibody | Rabbit anti-BLBP | Sigma | Cat. #ABN14; RRID:AB_10000325 | 1:1000 |
| Antibody | Rabbit anti-PH3 | Millipore | Cat. #06–570; RRID:AB_310177 | 1:2000 |
| Antibody | Rabbit anti-sox2 | Abcam | Cat. #ab97959; RRID:AB_2341193 | 1:500 |
| Antibody | Alexa Fluor 647 goat anti-mouse IgG(H+L) | ThermoFisher | Cat. #A-21235; RRID:AB_2535804 | 1:1000 |
| Sequence-based reagent | Cre-F | This paper | PCR primers | 5'- ATGTCCAATCTTCTAACCGT-3' |
| Sequence-based reagent | Cre-R | This paper | PCR primers | 5'- TTAGTCTCCATCCTCCAGCA-3' |
| Sequence-based reagent | Foxd3-F | This paper | PCR primers | 5'-CAGGGATCCATGA CCCTGTCTGGAGGCA-3' |
| Sequence-based reagent | Foxd3-R | This paper | PCR primers | 5'-GAACTCGAG TCATTGA GAAGGCCATTTCGATA-3' |
| Sequence-based reagent | Sox2-F | This paper | PCR primers | 5'-GTGAGTAACTTTT GGGTGTGCGG-3' |
| Sequence-based reagent | Sox2-R | This paper | PCR primers | 5'-TTAAACCGATTTTC TCGAAAGTCTAC-3' |
| Sequence-based reagent | Mmp17b-F | This paper | PCR primers | 5'-GGGAAGTGCTG TGGATGTTT-3' |
| Sequence-based reagent | Mmp17b-R | This paper | PCR primers | 5'-TAATACGACTCACTATAGATG AAACTCGAGCAGTGTTGG-3' |
| Sequence-based reagent | Adamts3-F | This paper | PCR primers | 5'-TCCTGGGGCT AGACATGTGA-3' |

*Continued on next page*

*Continued*

| Reagent type (species) or resource | Designation | Source or reference | Identifiers | Additional information |
|---|---|---|---|---|
| Sequence-based reagent | Adamts3-R | This paper | PCR primers | 5'-TAATACGACTCACTATAGAGC GCACAGTACGGATTTGA-3' |
| Software | ImageJ/Fiji | ImageJ.nih.gov | RRID:SCR_003070 | |
| Software | Prism 9 | GraphPad softwares | RRID:SCR_002798 | |
| Software | Metamorph | Molecular Devices | RRID:SCR_002368 | |
| Software | Imaris 9.6 | Oxford Instruments | RRID:SCR_007370 | |
| Software | RStudio | RStudio | RRID:SCR_000432 | |

## Zebrafish husbandry

All animal studies were approved by the University of Virginia Institutional Animal Care and Use Committee. Zebrafish strains used in this study were: AB*, *neuregulin1*[z26] (*Perlin et al., 2011*), *erbb3b*[st48] (*Lyons et al., 2005*), *plexinA3*[p13umal] (*Palaisa and Granato, 2007*), *Gt(foxd3:mcherry)*[ct110R] (*Hochgreb-Hägele and Bronner, 2013*), *Tg(XlTubb:dsred)*[zf148] (*Peri and Nüsslein-Volhard, 2008*), *Tg(sox10(4.9):nls-Eos)*[w18], *Tg(sox10(4.9):eos)*[w9] (*McGraw et al., 2012*), *Tg(olig2:dsred2)*[vu19], *Tg(olig2:egfp)*[vu12] (*Shin et al., 2003*), *Tg(nkx2.2a:megfp)*[vu17] (*Kucenas et al., 2008b*), *Tg(mbp:egfp-caax)*[ue2] (*Almeida et al., 2011*), *Tg(neuroD:gal4)*[uva22] (*Fontenas et al., 2019*), *Tg(sox10(7.2): mrfp)*[vu234] (*Kucenas et al., 2008b*), *Tg(sox10(4.9):tagrfp)*[uva5], *Tg(nkx2.2a(3.5):nls-mcherry)*[uva2], *Tg(nkx2.2a(3.5):nls-egfp)*[uva1], *Tg(ubi:zebrabow-m)*[a131] (*Pan et al., 2013*), *Tg(UAS:hNrg1typeIII)*[st85] (*Perlin et al., 2011*), *Tg(hb9:yfp-ntr)*[lmc008] (*Mathias et al., 2014*), *Tg(nkx2.2a(3.5):mcerulean3)*[uva41], *Tg(nkx2.2a(3.5):cre)*[uva42], and *Tg(sox2(2.9) :egfp)*[uva55]. *Table 1* denotes abbreviations used for each mutant and transgenic line. Embryos were raised at 28.5°C in egg water and staged by hours or days post fertilization (hpf and dpf, respectively). Embryos of either sex were used for all experiments (*Kimmel et al., 1995*). Phenyl-thiourea (PTU) (0.004%) in egg water was used to reduce pigmentation for imaging. Stable, germline transgenic lines were used in all experiments.

## Generation of transgenic lines

All constructs were generated using the Tol2kit Gateway-based cloning system (*Kwan et al., 2007*). Vectors used for making the expression constructs were p5E-nkx2.2a(−3.5) (*Pauls et al., 2007*), pME-mcerulean3 (*Zhu et al., 2019*), pME-Cre, p5E-sox2(−2.9), as well as pME-nls-eGFP, p3E-polyA, pDestTol2pA2 and pDestTol2CG2 destination vectors (*Kwan et al., 2007*).

To build pME-Cre, we amplified cre coding sequences from pCS2-Cre.zf1 (*Horstick et al., 2015*) using the following primers: forward primer, 5'- ATGTCCAATCTTCTAACCGT-3' and reverse primer, 5'- TTAGTCTCCATCCTCCAGCA-3'. The resulting PCR product was subcloned into pCR8/GW/ TOPO (Invitrogen) to generate a pME vector for Gateway cloning. pCS2-Cre.zf1 was a gift from Harold Burgess (Addgene plasmid #61391).

To generate the *Tg(sox2:egfp)* line, we amplified 2.9 kb of sequence immediately upstream of the sox2 gene (NM_213118) from wild-type genomic DNA using the following primers: forward primer, 5'-GTGAGTAACTTTTGGGTGTGCGG-3' and reverse primer, 5'-TTAAACCGATTTTC TCGAAAGTCTAC-3'. The resulting PCR fragment was subcloned into pENTR 5´-TOPO (Invitrogen) to generate a p5E entry for Gateway cloning. p5E, pME and p3E-polyA vectors were ligated into destination vectors through LR reactions (*Ashton et al., 2012*). Final constructs were amplified, verified by restriction digests, and sequenced to confirm the insertions. To generate stable transgenic lines, plasmid DNAs were microinjected at a concentration of 25 ng/µL in combination with 10 ng/µL *Tol2* transposase mRNA at the one-cell stage and screened for founders (*Kawakami, 2004*).

## Generation of synthetic mRNA

For *foxd3* overexpression experiments, we amplified *foxd3* coding sequences using the following primers (forward primer, 5'-CAGGGATCCATGACCCTGTCTGGAGGCA-3', reverse primer, 5'-GAAC TCGAG TCATTGAGAAGGCCATTTCGATA-3') and subcloned the PCR fragment into pCS2+ with

**Table 1.** Strains and transgenic lines.

Picture description: Cross section of the spinal cord showing Sox2[+] neural precursors (magenta), *nkx2.2a*[+] precursors (cyan) and Sox10[+] glia (yellow) at 4 days post-fertilization.

| Full name | abbreviation | Reference |
|---|---|---|
| *Tg(mbp:eGFP-CAAX)*[ue2] | mbp:egfp-caax | **Almeida et al., 2011** |
| *Gt(foxd3:mcherry)*[ct110R] | foxd3:mcherry | **Hochgreb-Hägele and Bronner, 2013** |
| *Tg(XlTubb:DsRed)*[zf148] | nbt:dsred | **Peri and Nüsslein-Volhard, 2008** |
| *Tg(neuroD1:Gal4; cmlc2:eGFP)*[uva22] | neuroD:Gal4 | **Fontenas et al., 2019** |
| *Tg(nkx2.2a:meGFP)*[vu17] | nkx2.2a:megfp | **Kucenas et al., 2008b** |
| *Tg(nkx2.2a(3.5):Cre; cmlc2:eGFP)*[uva42] | nkx2.2a:cre | This paper |
| *Tg(nkx2.2a(3.5):mCerulean3)*[uva41] | nkx2.2a:mcerulean | This paper |
| *Tg(nkx2.2a(3.5):nls-eGFP)*[uva1] | nkx2.2a:nls-egfp | This paper |
| *Tg(nkx2.2a(3.5):nls-mCherry)*[uva2] | nkx2.2a:nls-mcherry | **Zhu et al., 2019** |
| *Tg(olig2:eGFP)*[vu12] | olig2:egfp | **Shin et al., 2003** |
| *Tg(olig2:DsRed2)*[vu19] | olig2:dsred | **Shin et al., 2003** |
| *Tg(sox10(4.9):Eos)*[w9] | sox10:eos | **McGraw et al., 2012** |
| *Tg(sox10(4.9):nls-Eos)*[w18] | sox10:nls-eos | **McGraw et al., 2012** |
| *Tg(sox10(4.9):TagRFP)*[uva5] | sox10:tagrfp | **Zhu et al., 2019** |
| *Tg(sox10(7.2):mRFP)*[vu234] | sox10:mrfp | **Kucenas et al., 2008b** |
| *Tg(UAS:hNrg1 type III)* | UAS:hNrg1 | **Perlin et al., 2011** |
| *Tg(ubi:Zebrabow-M)*[a131] | ubi:zebrabow | **Pan et al., 2013** |
| *Tg(2xNRSE-2xMnx1-Mmu.Fos:KalTA4,5xUAS-ADV.E1b:GAP-YFP-2A-Eco.NfsBT41Q/N71S/F124T)*[lmc008] | hb9:yfp-ntr | **Mathias et al., 2014** |
| *Tg(sox2(2.9):eGFP)*[uva55] | sox2:egfp | This paper |
| AB* | wildtype | |
| *neuregulin1*[z26] | nrg1 | **Perlin et al., 2011** |
| *plexinA3* [p13umal] *(sidetracked)* | plexinA3 | **Palaisa and Granato, 2007** |
| *erbb3b*[st48] | erbb3b | **Lyons et al., 2005** |

BamH1 and Xho1. The vector was linearized with Not1 and *foxd3* mRNA was transcribed using mMESSAGE mMACHINE sp6 transcription kit (Ambion). For *adamts3* overexpression experiments, we linearized the pCR-XL-TOPO vector containing human ADAMTS3 full length cDNA (genomics-online.com ABIN3996515) with BamH1 and human *adamts3* mRNA was transcribed using mMESSAGE mMACHINE t7 transcription kit (Ambion).

## In vivo imaging

Embryos were anesthetized with 0.01% 3-aminobenzoic acid ester (Tricaine), immersed in 0.8% low-melting point agarose and mounted in glass-bottomed 35 mm Petri dishes (Electron Microscopy Sciences). After mounting, the Petri dish was filled with egg water containing PTU and Tricaine. A 40X water objective (NA = 1.1) mounted on a motorized Zeiss AxioObserver Z1 microscope equipped with a Quorum WaveFX-XI (Quorum Technologies) or Andor CSU-W1 (Oxford Instruments) spinning disc confocal system was used to capture all images. Images were imported into Metamorph (Molecular devices), Image J or Imaris (Oxford Instruments). Images of *foxd3:mcherry* embryos were deconvolved using the Image J plugin Iterative Deconvolve 3D to reduce background signal. All images were then imported into Adobe Photoshop. Adjustments were limited to levels, contrast, and cropping.

## Sectioning

Embryos were fixed in 4% paraformaldehyde (PFA) at 4℃ overnight. After fixation, embryos were embedded in 1.5% agarose/30% sucrose and frozen in 2-methylbutane chilled by immersion in liquid nitrogen. We collected 20 μm transverse sections on microscope slides using a cryostat microtome. Sections were covered with DAPI fluoromount-G (Southern Biotech). A 40X water objective (NA = 1.1) or a 63X water objective (NA = 1.2) mounted on a motorized Zeiss AxioObserver Z1 microscope equipped with a Quorum WaveFX-XI (Quorum Technologies) or Andor CSU-W1 (Andor Oxford Instruments) spinning disc confocal system was used to capture all images. Images were imported into Image J and Adobe Photoshop. Adjustments were limited to levels, contrast, and cropping.

## Eos photoconversion

For whole-embryo Eos photoconversion, *Tg(sox10:eos)* embryos were mounted for imaging as described above and then exposed to UV light using a DAPI filter for 1 min with a Zeiss Axiozoom V16 microscope at 48 hpf.

## Wholemount immunohistochemistry

Dechorionated embryos were fixed with 4% PFA overnight at 4℃, washed in PBSTx (1% Triton X-100, 1x PBS) for 5 min, permeabilized in acetone at room temperature for 5 min and at −20℃ for 10 min, followed by a 5 min wash with PBSTx. Embryos were then blocked in 5% goat serum/PBSTx for 1 hr, incubated in primary antibody with 5% goat serum/PBSTx for 1 hr at room temperature and overnight at 4℃. Embryos were washed extensively with PBSTx at room temperature and incubated in secondary antibody overnight at 4℃. After antibody incubation, embryos were washed extensively with PBSTx and stored in PBS at 4℃ until imaging. The following antibodies were used: rabbit anti-Sox10 (1:5000) (*Binari et al., 2013*), chicken anti-GFP (1:500; abcam), Alexa Fluor 488 goat anti-chicken (1:1000; ThermoFisher) and Alexa Fluor 647 goat anti-rabbit IgG(H+L) (1:1000; Thermo-Fisher). Embryos were mounted in glass-bottomed Petri dishes for imaging as described above.

## Immunohistochemistry on section

Embryos were fixed in 4% PFA overnight at 4℃ and sectioned as described above. Sections were rehydrated in PBS for 30 min and blocked with 5% goat serum/PBS for 1 hr at room temperature. Primary antibody incubation was done overnight at 4℃. Secondary antibody incubation was done for 2 hr at room temperature. Antibodies used were: chicken anti-GFP (1:500; abcam ab13970), mouse anti-GFAP (1:1000; ZIRC zrf-1), rabbit anti-PH3 (1:2000; Millipore 06–570), rabbit anti-BLBP (1:1000; Millipore ABN14), rabbit anti-Sox2 (1:500; abcam ab97959), Alexa Fluor 647 goat anti-rabbit IgG(H+L) (1:1000), Alexa Fluor 488 goat anti-chicken IgG(H+L) (1:1000), Alexa Fluor 647 goat anti-mouse IgG(H+L) (1:1000).

## In situ hybridization

Larvae were fixed in 4% PFA at 4℃ overnight and stored in 100% methanol at −20℃ and processed for in situ RNA hybridization as described previously (*Hauptmann and Gerster, 2000*). Plasmids were linearized with appropriate restriction enzymes and cRNA preparation was carried out using Roche DIG-labeling reagents and RNA polymerases (NEB). We used previously published probes for *olig2* (*Park et al., 2002*), *plp1a* (*Brösamle and Halpern, 2002*), *krox20* (*Monk et al., 2009*), and *wif1* (*Smith et al., 2014*). *Mmp17b* and *adamts3* probes were generated using the following primers (*mmp17b* forward primer, 5'-GGGAAGTGCTGTGGATGTTT-3', reverse primer, 5'-TAATACGAC TCACTATAGATGAAACTCGAGCAGTGTTGG-3'; *adamts3* forward primer, 5'-TCCTGGGGC TAGACATGTGA-3', reverse primer, 5'-TAATACGACTCACTATAGAGCGCACAGTACGGATTTGA-3') and t7 RNA polymerase. Sectioning was performed as described above and sections were covered with Aqua-poly/mount (Polysciences). Images were obtained using a Zeiss AxioObserver inverted microscope equipped with Zen, using a 40x oil immersion objective. All images were imported into Adobe Photoshop. Adjustments were limited to levels, contrast, and cropping.

## Chemical treatments

For motor neuron ablation, dechorionated *hb9:yfp-ntr* embryos were treated with 20 mM metronidazole (Sigma)/2% DMSO in egg water from 9 to 24 hpf and 20 mM metronidazole/2% DMSO in PTU egg water from 24 to 72 hpf. Control siblings were treated with 2% DMSO in egg water or PTU egg water according to their stage.

For matrix metalloproteinase inhibition experiments, dechorionated *foxd3:mcherry;nkx2.2a:megfp* embryos were treated with 100 μM GM6001 (Enzo Life Sciences)/1% DMSO in PTU egg water from 36 to 72 hpf. Fresh drug was replaced prior to imaging at 48 hpf. Control siblings were treated with 1% DMSO in PTU egg water.

For cell cycle arrest experiments, dechorionated *foxd3:mcherry;nkx2.2a:nls-egfp* embryos were treated with 150 μM aphidicolin (Sigma) and 20 mM hydroxyurea (Sigma)/2% DMSO in PTU egg water from 48 to 72 hpf. Control siblings were treated with 2% DMSO in PTU egg water.

For the EdU incorporation assay, *nkx2.2a:nls-egfp* embryos were treated with 0.1 mM EdU/1% DMSO from 50 to 56 hpf in PTU egg water at 28.5℃, switched to PTU egg water at 56 hpf and fixed for 1 hr in 4% PFA at 72 hpf. Whole embryos were permeabilized with cold acetone for 7 min at −20℃ and stained for EdU using the Click-it EdU Cell Proliferation kit for Imaging with Alexa Fluor 647 dye (ThermoFisher), as detailed in the kit protocol. Click-it reaction was performed for 1 hr at room temperature followed by an immunostaining against Sox10 and GFP as described above.

## Genotyping

Genomic DNA was extracted using HotSHOT (hot sodium hydroxide and tris) (*Truett et al., 2000*). The primers used for genotyping *UAS:hNrg1typeIII*, *erbb3b*$^{st48}$, *nrg1*$^{z26}$ and *plexinA3*$^{p13umal}$ have previously been published (*Lyons et al., 2005*; *Palaisa and Granato, 2007*; *Perlin et al., 2011*).

## Quantification and statistical analysis

### Fluorescence intensity measurement

For MEP glial eGFP fluorescence intensity analysis, we measured the mean gray value of four MEP glia, four oligodendrocytes, or four neural tube cells per image and the mean gray value of four regions of background using Image J, and analyzed seven images from seven embryos per transgenic line. We calculated the corrected cell fluorescence intensity by subtracting the background mean gray value.

For the time course analysis of MEP glia eGFP fluorescence intensity, we measured the mean eGFP gray value of *foxd3:mcherry*$^+$/*olig2:egfp*$^+$ and *foxd3:mcherry*$^+$/*nkx2.2a:nls-egfp*$^+$ MEP glia at 48, 50, 55, and 72 hpf using single z-planes of confocal images. To avoid photobleaching, different groups of embryos from the same clutch were used for each timepoint assessed. For each measurement, the region of interest was defined using the mCherry channel to better visualize MEP glia and measurements were done using the same coordinates in the corresponding GFP channel. For *olig2* and *nkx2.2a*, the mean gray value measured at 48 hpf was normalized to 1 and all mean gray values from the 50, 55, and 72 hpf timepoints were compared to that value. The number of MEP glia used for this analysis is as follows (*olig2*, 48 hpf: n = 27, 50 hpf: n = 27, 55 hpf: n = 24, 72 hpf: n = 20; and *nkx2.2a*, 48 hpf: n = 13, 50 hpf: n = 26, 55 hpf: n = 25, 72 hpf: n = 31).

For Zebrabow intensity profile analysis, we split the image channels and measured the mean gray value of each individual MEP glia in the red channel, green channel, and blue channel, in Image J. Individual MEP glia were identified by their x, y, and z coordinates. Percentages of fluorescence of reach channel was calculated by dividing the mean gray value by the maximum intensity of each image using the plugin Histogram. The zebrabow intensity profile ternary plot was made in R studio with the Ternary package.

### Quantification of MEP glia and OPC numbers

Student t tests and for multiple comparisons, one-way ANOVAs followed by Tukey's multiple comparison tests, were performed using Prism 8. For violin plots, dotted lines represent quartiles and the central dashed line marks the median. The data in plots and the text are presented as means ± SEM.

## Acknowledgements

We thank Lori Tocke for zebrafish care and members of the Kucenas lab for valuable discussions. We thank Drs. Yunlu Zhu, Eyleen O'Rourke, and Sarah Siegrist for helpful suggestions, and Dr. William Talbot for the *Tg(UAS:hNrg1typeIII)* line. This work was funded by NIH/National Institute of Neurological Disorders and Stroke (NINDS): NS072212 (SK) and NS107525 (SK).

## Additional information

### Funding

| Funder | Grant reference number | Author |
|---|---|---|
| NINDS | NS072212 | Sarah Kucenas |
| NINDS | NS107525 | Sarah Kucenas |

The funders had no role in study design, data collection and interpretation, or the decision to submit the work for publication.

### Author contributions

Laura Fontenas, Conceptualization, Resources, Data curation, Formal analysis, Validation, Investigation, Visualization, Methodology, Writing - original draft, Project administration, Writing - review and editing; Sarah Kucenas, Conceptualization, Supervision, Funding acquisition, Validation, Writing - original draft, Project administration, Writing - review and editing

### Author ORCIDs

Laura Fontenas https://orcid.org/0000-0003-0544-0147
Sarah Kucenas https://orcid.org/0000-0002-1950-751X

### Ethics

Animal experimentation: All animal studies were approved by the University of Virginia Institutional Animal Care and Use Committee, Protocol #3782.

### Decision letter and Author response

Decision letter https://doi.org/10.7554/eLife.64267.sa1
Author response https://doi.org/10.7554/eLife.64267.sa2

## Additional files

### Supplementary files

• Transparent reporting form

### Data availability

All data generated or analyzed during this study are included in the manuscript and supporting files.

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
