## [Decision Letter]

**Acceptance summary:**

This study furthers our molecular understanding about the CNS origins and development of a novel peripheral glial cell type, the motor exit point (MEP) glia, in zebrafish. The authors demonstrate that MEP glia require a foxd3-MMP pathway, as well as signals from motor axon nerves, for their exit from the CNS into the periphery and for their proper development. The authors also show that MEP glia originates from the lateral floor plate radial glial precursors and not from the pMN domain as initially described. This study fills a critical gap in our knowledge about the early formation of this novel cell type in vivo and provides a significant advance.

**Decision letter after peer review:**

Thank you for submitting your article "Spinal cord precursors utilize neural crest cell mechanisms to generate hybrid peripheral myelinating glia" for consideration by *eLife*. Your article has been reviewed by three peer reviewers, one of whom is a member of our Board of Reviewing Editors, and the evaluation has been overseen by Marianne Bronner as the Senior Editor. The following individuals involved in review of your submission have agreed to reveal their identity: Tatiana Solovieva (Reviewer #2); Rosa a Uribe (Reviewer #3).

The reviewers have discussed the reviews with one another and the Reviewing Editor has drafted this decision to help you prepare a revised submission.

Summary:

The manuscript provides a detailed and beautiful characterization of previously described neural tube derived glia that migrate out into the periphery along motor axons (MEP, motor exit point glia).

The authors performed an expression analysis using a combinations of reporter and brainbow lines and determined that MEP glia are *foxd3^+^/sox10^+^/olig2^+^/nkx2.2a^+^/mbp^+^* neural tube-derived, peripheral myelinating glia that express a unique combination of central and peripheral markers, and identify *nkx2.2a* as a novel MEP glial marker. In addition, while there are many studies describing the roles of foxd3 in peripheral tissue development, none have described a role for this gene in CNS glial development. The authors also show that MEP glia originates from the lateral floor plate radial glial precursors and not from the pMN domain as they initially described. The also identify signals that instruct MEP glia to delaminate from the ventral neural tube to migrate through MEP TZs and onto peripheral motor axons. For example, they demonstrate that MEP glial development depends on axonally derived neuregulin1 and overexpression of axonal cues induces additional MEPs to form. The experiments are carefully done and follow a logic progression. However, some interpretations need to be clarified, additional in situs need to be performed and several of their findings should be followed up to provide more mechanistic insight.

Essential revisions:

1) The authors state that foxd3 is required for delamination of MEPs and not their specification. One downstream candidate target of foxd3 causing is the MMP adamts3. This is an interesting and novel finding and should be followed up. Does overexpression of adamts13 induce delamination of more MEPs? In embryos in which foxd3 is overexpressed and that show extra MEPs, is adamts13 upregulated? Is there anything known about adamts3 in glial development? Is adamts3 expression also absent in other zebrafish mutants that display MEP glial exit phenotypes the authors have examined, or is this particular to foxd3-/- embryos?

2) Figure 1G-I and Figure 1—figure supplement 1D. The authors state that olig2 is weaker expressed than *nkx2.2a* in MEPs. This conclusion is based on the analysis of transgenic reporter line, which may not completely mimic endogenous expression. Additionally, nuclear localized GFP may be more stable than cytoplasmic GFP making a direct comparison based on fluorescence difficult. The authors should perform in situs for nkx2.2a. It would be informative to know whether MEP glia continue to express nkx2.2a transcript during their exit from the spinal cord and migration along axons. In addition, when looking at the expression levels of olig2 in MEPs prior to their exit from the CNS, the authors should look only at MEP precursors (e.g. *Olig2^+^ve, Nkx2.2^+^ve, Foxd3^+^ve*, as in Figure 2C) and not just any Olig2 positive cells in the pMN domain (as appears in Fig1G-H). This is important because there appears to be a difference in Olig2 levels between MEP precursors and other *Olig2^+^*ve cells in the pMN domain as evident in Figure 2—figure supplement 1A and Figure 2A and Figure 2C. Based on the images available, could it be possible that the level of Olig2 is relatively low in MEPs compared to their neighbouring cells in the pMN domain, and that this level doesn't change much after migration out of the CNS?

3) Figure 7. The authors should show if *nrg1-3* is expressed in motor neurons or reference the Lee., 2020paper. They should also examine if erbbs are expressed in MEPs. It is stated in subsection “MEP glial development depends on axonal Neuregulin 1 type III” that previous work showed that *erbb3b* is expressed in MEPs, but I could not find that data. Those references show that *erbb3b* mutants have MEP phenotypes, but do not show in which cells erbb3b is acting. The authors should perform *erbb3b* in situ analyses during MEP formation and migration.

4) Figure 6—figure supplement 1A-B. The photo conversion experiment is not described in the main text. At what time point were the cells converted? There appear to be unconverted Schwann cells along DRG axons, illustrating that the authors possibly cannot distinguish between Schwann cells and MEPs in this experiment?

5) Figure 8. The authors show that overexpression of nrg1 increases proliferation of *nkx2.2a* (+) cells in the ventral spinal cord which may not be MEP progenitors. The authors should study proliferation of nkx2.2a (+) cells at a time point where they can distinguish MEPs from their progenitors (along the motor root). Are the increased numbers of *nkx2.2a* (+) cells seen in Figure 8B due to increased numbers of MEPs exiting the spinal cord, or due to subsequent increased proliferation or both? Is proliferation decreased in nrg1 mutants? The authors should do the EdU assay in wildtype, nrg1 mutants and nrg1 over expression at a time point where they can distinguish MEPs from their progenitors.

6) With regard to the MMP inhibitor experiments, it would be useful for assessment of the MEP phenotype to include the time lapse videos of the GM6001-treated and DMSO-treated *nkx2.2a:megfp;foxd3:mcherry* embryos.

---

## [Author Response]

Essential revisions:1) The authors state that foxd3 is required for delamination of MEPs and not their specification. One downstream candidate target of foxd3 causing is the MMP adamts3. This is an interesting and novel finding and should be followed up. Does overexpression of adamts13 induce delamination of more MEPs? In embryos in which foxd3 is overexpressed and that show extra MEPs, is adamts13 upregulated? Is there anything known about adamts3 in glial development? Is adamts3 expression also absent in other zebrafish mutants that display MEP glial exit phenotypes the authors have examined, or is this particular to foxd3-/- embryos?

To test whether overexpression of adamts3 induces delamination of more MEP glia, we overexpressed *adamts3* by injecting human *adamts3* mRNA at the one-cell stage into *foxd3:mcherry;nkx2.2a:nls-egfp* embryos. We then used time-lapse imaging between 48 and 72 hpf and analyzed the number of MEP glia delaminating from the lateral floor plate. In this experiment, we observed no difference between control and *adamts3* overexpressing larvae. In both scenarios, we observed just one MEP glia delaminate in each somite. This result is not surprising as *adamts3* overexpression might result in a greater degradation of the basal lamina that surrounds the neural tube. However, we hypothesize from our other findings in the paper that *adamts3* functions downstream of the transcription factor *foxd3* and therefore, might not play a role on MEP glial specification in the precursor domain. We have added a sentence in the text to reflect this data.

(control: mean ± SEM = 1.06 ± 0.056, n=18 nerves from 5 embryos; *adamts3* mRNA: mean ± SEM = 1 ± 0.00, n=29 nerves from 9 embryos; p=0.38).We also attempted to approach this question by trying to knock down *adamts3* with CRISPR/Cas9. We tested 9 gRNAs (designed with CHOPCHOP and crisprscan.org), but unfortunately, were unable to identify a gRNA that cuts, and to our knowledge, there is no validated *adamts3* mutant available.

To determine if *adamts3* is upregulated in larvae overexpressing *foxd3,* we performed an *adamts3 in situ* hybridization in control (n=30 embryos) and *foxd3* mRNA-injected embryos (n=30 embryos) and did not observe any significant change of expression of *adamts3* when compared to control larvae at 48 hpf.

**Author response image 2. respfig2:** 

Additionally, to our knowledge, there is no literature on *adamts3* in glial development and future studies will be needed to explore this mechanism.Finally, unfortunately, there are no other known mutants that share a phenotype of a failure of MEP glial exit. Several mutants affect MEP glial migration once in the periphery, including *erbb3b*, *nrg1,* and *colourless* (*sox10*). We are obviously very interested in identifying genes that cause a failure of exit from the CNS, and we are currently conducting a forward mutagenesis screen to do just this. However, we are still screening our F3 families and any identified mutants will be part of future manuscripts.

2) Figure 1G-I and Figure 1—figure supplement 1D. The authors state that olig2 is weaker expressed than nkx2.2a in MEPs. This conclusion is based on the analysis of transgenic reporter line, which may not completely mimic endogenous expression. Additionally, nuclear localized GFP may be more stable than cytoplasmic GFP making a direct comparison based on fluorescence difficult. The authors should perform in situs for nkx2.2a. It would be informative to know whether MEP glia continue to express nkx2.2a transcript during their exit from the spinal cord and migration along axons. In addition, when looking at the expression levels of olig2 in MEPs prior to their exit from the CNS, the authors should look only at MEP precursors (e.g. Olig2^+^ve, Nkx2.2^+^ve, Foxd3^+^ve, as in Figure 2C) and not just any Olig2 positive cells in the pMN domain (as appears in Fig1G-H). This is important because there appears to be a difference in Olig2 levels between MEP precursors and other Olig2^+^ve cells in the pMN domain as evident in Figure 2—figure supplement 1A and Figure 2A and Figure 2C. Based on the images available, could it be possible that the level of Olig2 is relatively low in MEPs compared to their neighbouring cells in the pMN domain, and that this level doesn't change much after migration out of the CNS?

*olig2:egfp* is a BAC transgenic line (Park et al., 2007). Therefore, we are confident that its expression faithfully mimics endogenous expression.

We respectfully disagree that a *nkx2.2a in situ* would inform us as to whether MEP glia continue to express *nkx2.2a* transcript during their exit and migration along axons. As the reviewers noted below in comment 16, *nkx2.2a* is also expressed in perineurial glia along the same axons, and an *in situ* would not give us the cellular resolution to discern between these two glial populations. However, we have conducted studies using a *nkx2.2a:megfp* BAC transgenic line, in combination with other lines, that mimics faithful endogenous expression of *nkx2.2a* (Kucenas et al., 2008) to trace MEP glia, and we have detected *nkx2.2a* expression in MEP glia from 45 to 72 hpf (Figure 4A) and 4 dpf (Figure 1E). Because GFP expressed by this transgene is detected at 4 dpf in MEP glia, 48 hours after they have excited the CNS, we conclude that they continue to express *nkx2.2a* well after their exit.

We agree with the reviewer’s point about investigating MEP glial expression of *olig2* before they exit the CNS and have repeated our fluorescence intensity experiment and measured GFP intensity of *foxd3^+^/olig2^+^* MEP glia and *foxd3^+^/nkx2.2a^+^* MEP glia in the spinal cord at 48 hpf before they exit, when they exit the spinal cord at 50 hpf, and in the periphery at 55 hpf and 72 hpf using single z-plane confocal images. We used different embryos from the same clutch for each time point we assessed to prevent photobleaching and to preserve fluorescence. Our analysis shows that while *nkx2.2a* levels remain quite stable over time in MEP glia, those of *olig2* steadily decrease by 65% between 48 and 72 hpf. These results are now included in Figure 1—figure supplement 2A-C.

3) Figure 7. The authors should show if nrg1-3 is expressed in motor neurons or reference the Lee., 2020paper. They should also examine if erbbs are expressed in MEPs. It is stated in subsection “MEP glial development depends on axonal Neuregulin 1 type III” that previous work showed that erbb3b is expressed in MEPs, but I could not find that data. Those references show that erbb3b mutants have MEP phenotypes, but do not show in which cells erbb3b is acting. The authors should perform erbb3b in situ analyses during MEP formation and migration.

We have added references to several studies showing *nrg1 type 3* is expressed in motor neurons, including the Glia paper of Lee at al., 2020.

To demonstrate that MEP glia are directly affected in *erbb3b* mutants, we performed a *wif1in situ* hybridization in *erbb3b* mutants at 72 hpf. In these studies, we observed that the MEP glial phenotype phenocopies what we observed in *nrg1* mutant larvae, where MEP glia migrate away from the motor root, demonstrating that both *nrg1* and its receptor *erbb3b* regulate MEP glial migration. We have included these results in Figure 7C.

Regarding MEP glial expression of *erbb3b –* this has been shown previously, which is why we did not include it in our manuscript. See Langworthy and Appel, 2012 for an *in-situ* hybridization showing continuous expression of *erbb3b* all the way down the motor nerves (mn) starting from the motor exit point. This data shows strong *erbb3b* expression at the motor roots, where MEP glia, but not Schwann cells, are located.

4) Figure 6—figure supplement 1A-B. The photo conversion experiment is not described in the main text. At what time point were the cells converted? There appear to be unconverted Schwann cells along DRG axons, illustrating that the authors possibly cannot distinguish between Schwann cells and MEPs in this experiment?

We have added the following photoconversion description “UV light-induced photoconversion of *sox10:eos* embryos at 48 hpf allows us to distinguish red, photoconverted neural crest-derived Schwann cells from green, non-photoconverted MEP glia and OL lineage cells, as previously described (Smith et al., 2014)” in the text corresponding to Figure 6—figure supplement 1A-B.

However, we disagree and do not think there are any unphotoconverted Schwann cells along DRG axons in Figure 6—figure supplement 1B. In Figure 6—figure supplement 1A, which is a lower resolution/magnification but shows more ectopic motor exit points than 1B, the green channel might have been set too bright and masks some of the existing red signal. For this reason, we have now readjusted the brightness of the green and red channels in Figure 6—figure supplement 1A to allow for clearer visualization of Schwann cells vs MEP glia.

5) Figure 8. The authors show that overexpression of nrg1 increases proliferation of nkx2.2a (+) cells in the ventral spinal cord which may not be MEP progenitors. The authors should study proliferation of nkx2.2a (+) cells at a time point where they can distinguish MEPs from their progenitors (along the motor root). Are the increased numbers of nkx2.2a (+) cells seen in Figure 8B due to increased numbers of MEPs exiting the spinal cord, or due to subsequent increased proliferation or both? Is proliferation decreased in nrg1 mutants? The authors should do the EdU assay in wildtype, nrg1 mutants and nrg1 over expression at a time point where they can distinguish MEPs from their progenitors.

To investigate MEP glial proliferation once in the periphery, we performed EdU labeling in wildtype and *nrg1* overexpressing larvae. To do this, we treated *nkx2.2a:nls-egfp* control and *neurod:gal4;uas:hNrg1* embryos with EdU from 50 to 56 hpf, when all MEP glia are in the periphery and actively proliferating. At 72 hpf, we fixed the larvae, performed the EdU reaction and an Sox10 immunostaining (as we previously did in the experiment presented in Figure 5—figure supplement 1). In this experiment, we observed an increase in the number of EdU^+^ MEP glia along motor nerves, as well as an in increase in the total number of MEP glia present along motor nerves. These results were added to the manuscript and are presented in Figure 8—figure supplement 2A-C.

Regarding the second part of this reviewer comment – In *nrg1* mutants, peripheral axons are devoid of all glia, consistent with other studies (Lee et al., 2020; Perlin et al., 2011). Therefore, performing an EdU proliferation assay on these mutants would reveal nothing because there are no glia to assay. Therefore, we did not do this experiment.

6) With regard to the MMP inhibitor experiments, it would be useful for assessment of the MEP phenotype to include the time lapse videos of the GM6001-treated and DMSO-treated nkx2.2a:megfp;foxd3:mcherry embryos.

We have included time-lapse videos of the DMSO-treated (Video 5) and GM6001-treated (Video 6) *nkx2.2a:megfp;foxd3:mcherry* embryos.